# Executing Arithmetic: Fine-Tuning Large Language Models as Turing Machines

## Abstract

Large Language Models (LLMs) have demonstrated remarkable capabilities across a wide range of natural language processing and reasoning tasks. However, their performance in the foundational domain of arithmetic remains unsatisfactory. When dealing with arithmetic tasks, LLMs often memorize specific examples rather than learning the underlying computational logic, limiting their ability to generalize to new problems. In this paper, we propose a Composable Arithmetic Execution Framework (CAEF) that enables LLMs to learn to execute step-by-step computations by emulating Turing Machines, thereby achieving a true mastery of computational logic. Moreover, the proposed framework is highly scalable, allowing composing learned operators to significantly reduce the difficulty of learning complex operators. In our evaluation, CAEF achieves nearly 100% accuracy across seven common mathematical operations on the LLaMA 3.1-8B model, effectively supporting computations involving operands with up to 100 digits, a level where GPT-4o falls short noticeably in some settings.

## 1 Introduction

Large Language Models (LLMs) have made significant strides in recent years, showcasing extraordinary capabilities across a range of natural language processing (NLP) tasks (Dubey et al., 2024; Jiang et al., 2024; Chowdhery et al., 2023), and in some cases, even surpassing human performance in specific benchmarks (Achiam et al., 2023). However, despite these advancements, LLMs still face significant challenges in performing arithmetic. Current research indicates that when presented with arithmetic problems, LLMs often rely on memorizing specific expressions and their corresponding outcomes rather than grasping the fundamental logic of arithmetic operations (Wu et al., 2023b). This inherent limitation poses a substantial barrier to their effective application in fields that demand essential computational skills.

To enhance the performance of LLMs in solving arithmetic problems, two primary approaches have been developed. The first approach positions the LLM as an agent that relies on an external calculator to perform computations (Hao et al., 2024; Ruan et al., 2023). In this setting, the LLM's role is limited to providing the operands and invoking the appropriate operations. Although this method effectively simplifies the challenge of arithmetic for LLMs, it misses the opportunity for the models to learn computational logic, preventing LLMs from comprehending the underlying principles of arithmetic. Given that arithmetic serves as the foundation of mathematics, the lack of arithmetic ability may significantly impede the LLM's capability to grasp more complex mathematical concepts. The second approach focuses on stimulating the LLM's intrinsic capabilities, employing prompt engineering or fine-tuning techniques to enable the model to master arithmetic computations and solve problems through reasoning (Kojima et al., 2022; Huang et al., 2022; Yu et al., 2023). This approach typically involves the LLMs generating intermediate steps before reaching a final result.

Although the second approach is promising, it faces two significant challenges. The first challenge is that, under simple supervised fine-tuning, LLMs tend to memorize examples from the training set (Hu et al., 2024). As the length of the operands increases, the sample space expands exponentially, making it impractical for the LLM to memorize all possible examples. To fundamentally overcome this limitation, LLMs should primarily *learn and execute computational logic*, mirroring how humans systematically master arithmetic, rather than relying on memorization.

Figure 1: An illustrative CAEF flowchart demonstrates the execution of the *Multiplication* operation for $89 \times 2$. The aligner converts the original arithmetic expression into a Turing Machine-like representation that the *Multiplication* executor can process. Acting as an executor composer, the *Multiplication* executor calls upon two basic executors, i.e., *Less_than* and *Addition*, to perform the actual computation. All the executors and the aligner are executed by the LLM.

The second challenge involves learning how to compose basic operators to build complex arithmetic operators. These complex operators are typically execution procedures that contain conditional statements ($if\text{-}then\text{-}else$) and iterative statements ($loop$), with the basic operators treated as function calls within these procedures. By doing this, LLMs could gradually learn more complex arithmetic operations by focusing on their execution logic and calling the existing operators as necessary.

Mastering the execution of arithmetic is fundamentally equivalent to modeling computation. One famous mathematical model of computation is the Turing machine, which is formally introduced by Alan Turing (Turing et al., 1936). If the LLM learns to execute computational logic by simulating executing a Turing machine based on its transition functions for each operator, it could solve arithmetic problems through a multi-query approach. This approach involves the LLM iteratively performing computations based on the current state and command, and then generating the next state and command.

In this paper, we propose a Composable Arithmetic Execution Framework (CAEF) for LLMs to solve arithmetic problems solely. Inspired by the Turing machine, CAEF aims to teach LLMs the computational logic, enabling them to *execute* the logic for specific arithmetic operators and *compose* arithmetic operators into more complex ones. CAEF has two key characteristics:

**Executing arithmetic**. As illustrated in Figure 1, CAEF employs a three-step procedure for each arithmetic operator, supported by two independent components within the LLM: the *executor* and the *aligner*. The executor, responsible for performing the actual computations, learns the underlying computational logic by modeling the transition function of the corresponding arithmetic Turing machine. This allows the LLM to iteratively generate intermediate results and ultimately produce the final output. The aligner serves as an interface, converting raw arithmetic expressions (e.g., $89 \times 2 =$) into a format that the executor can directly process. Once the executor completes its execution, the aligner transforms the executor's output back into the final result. In our framework, both the executor and the aligner are implemented as separate LoRA adapters (Hu et al., 2021).

**Composing operators**. Complex operators can often be composed of basic or simpler ones, hierarchically or recursively. In CAEF, we design an *executor composer* that is responsible for the high-level execution procedures of complex operators and allows function calls to invoke other pre-learned arithmetic operators. Since each operator is implemented as a LoRA adapter, function calls in CAEF are executed by automatically switching LoRA adapters, following the LLM's generated command. Thus, CAEF could facilitate the handling of more intricate computations.

Using the proposed framework, we have implemented seven operators: $+$, $-$, $\times$, $\div$, $>$, $<$, and $==$, along with two auxiliary operators (refer to Appendix A.4). Each of these operators is based on existing computational logic, such as the Turing machine or algorithms used in CPU design (e.g., the subtraction operator is modeled similarly to how modern CPUs handle the subtraction operation.). Our experiments show that CAEF achieves high accuracy across all seven operators when using the LLaMA 3.1-8B model (Dubey et al., 2024). Compared to GPT-4o, the LLM equipped with CAEF demonstrates minimal impact from changes in operand length, effectively supporting computations involving operands with up to 100 digits. The main contributions of this paper are as follows:

- We propose a framework CAEF enabling LLM learning to execute the computational logic of operators by imitating the execution of Turing machine. Also, CAEF can naturally support composing multiple learned operators for operators with complex logic.

- We implement executors and aligners for seven arithmetic operators based on the proposed framework. The executor is responsible for performing the step-by-step computations iteratively, while the aligner serves as an interface, facilitating the bidirectional conversion between the internal representation of the executor and the original representation.

- The extensive evaluation shows that CAEF outperforms the existing LLMs with seven classic arithmetic tasks. The proposed CAEF enables the LLM to achieve almost $100\%$ accuracy when operands are up to 100 digits.

## 2 APPROACH: FRAMEWORK DESIGN

### 2.1 PROBLEM STATEMENT

Computational logic is fundamental to arithmetic. To truly master arithmetic, the LLM should learn and execute the underlying computational logic of arithmetic operations rather than merely memorizing examples of arithmetic expressions. For scalability, the LLM should be capable of constructing new operators by combining existing operators. For example, after learning *Addition* operation, the LLM could construct *Multiplication* by learning the computational logic of repeated addition could achieve multiplication.

Therefore, we need a framework that enables LLM to model arithmetic operators by learning to execute their underlying computational logic. In the field of automata, the Turing machine provides a suitable framework for describing this logic. Following the examples (e.g., Turing machines introduced in Sipser (1996)), we could build a Turing machine for common arithmetic operations, which can be a reference to create adequate datasets of execution steps for LLM training. Furthermore, the Turing machine inherently supports the combination of multiple Turing machines, making it ideal for constructing complex operations from existing ones. By emulating Turing machines, LLM can be designed to integrate multiple models, enabling it to execute more intricate arithmetic tasks.

### 2.2 LLM EXECUTES AS TURNING MACHINE

A Turing machine can be formally defined as a septuple $T = (Q, \Sigma, \Gamma, b, q_0, F, \delta)$, where $Q$ is a finite set of states, $\Sigma \subseteq \Gamma$ is a finite alphabet for input, $\Gamma$ is a finite tape alphabet, $b \in \Gamma$ is the blank symbol, $q_0 \in Q$ is the initial state, $F \subseteq Q$ is a set of final states, and $\delta$ is the transition function. When a Turing machine is in a non-halting state, the next action is determined by both the current symbol on the tape and the machine's current state. In each action, the machine updates the symbol on the tape, transitions to a new state, and moves the tape head either left or right. This process repeats iteratively until the machine reaches a halting state, at which point the computation is complete, and the result is saved on the tape.

LLM is the generative model for text-based language, so how to transfer all information from a Turing machine to the LLM effectively is challenging. A tailored representation system is necessary for LLMs to accurately learn computational logic. To facilitate this transfer, the system must incorporate states analogous to those of the Turing machine, such as the machine state and tape state, to indicate the current status of the computation, in other words, the step in the execution process. Additionally, the system should include commands that specify the actions to be executed based on the current state to ensure correct transitions to the next state. Thus, CAEF provides a text-based representation $< s_i, c_i >$ that effectively represents the state $s_i$ and the command $c_i$ for each step $i$ in the computational logic. Then, the state transition function $f$ (e.g. LLM or LLM fine-tuned with LoRA adapters) could use this representation at step$_i$ as the input to generate the next representation at step$_{i+1}$ as following:

$$s_{i+1}, c_{i+1} = f(s_i, c_i) \tag{1}$$

By formulating the representation of both input and output for Equation 1, the LLM is enabled to perform computations in a manner similar to that of a Turing Machine by *executing* step-by-step transitions.

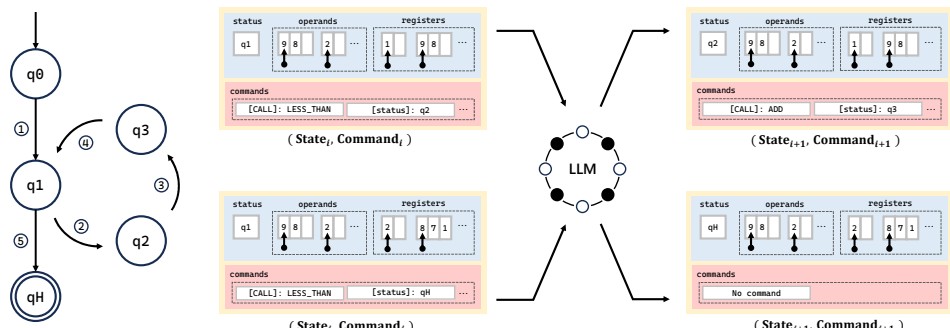

Figure 2: Diagram of the CAEF framework. The CAEF representation includes two required components: state and command, corresponding to areas ░░░ and ░░░ in the figure. The state part records the current status, operands, and registers that store intermediate variables and results, etc. The command consists of a set of actions, such as write operations and call operations. Upon receiving the state and command, the LLM generates the next state and the corresponding command, with each step corresponding to a transition in the state diagram on the left.

## 2.3 REPRESENTATION DESIGN

In this paper, we design a structured representation for arithmetic problems to enable the LLM to accurately execute computational logic. As illustrated in Figure 2, representation of the arithmetic problem includes: 1) a status indicating the current step of the computation, and 2) a "tape" that records all operands and essential information, such as the number of digits processed, any carryover during addition, and other intermediate results. To facilitate the LLM's learning of the execution process, the representation in CAEF explicitly includes the commands $c$ required for execution. These commands involve the *call* to the next *state* $s$ and other detailed actions, such as carrying over or moving the pointer. All the above elements are represented in text, which is friendly to LLM to deal with. Then, to make LLM execute based on the representation, CAEF structures the input as $< s_i, c_i >$ for current step$_i$, while the output is $< s_{i+1}, c_{i+1} >$ for the next step$_{i+1}$.

Besides modeling the representation, representation translation is another critical part of CAEF. In general, the original input of an arithmetic problem does not include the initial state or the first command to execute. Moreover, upon completing the computation, the raw output remains in the representation format. Thus, CAEF incorporates an aligner to manage the bidirectional translation between the original input/output and the representation. The aligner can also be implemented by fine-tuning a specific LoRA adapter. Notice that one key feature of the aligner should learn the ability to convert the left-to-right (L2R) representation of numbers into a right-to-left (R2L) format, as R2L is evaluated more effectively for LLM to operate the operands (Lee et al., 2023).

## 3 APPROACH: IMPLEMENTATION

Building on the conceptual design of CAEF, we present the detailed implementation of Equation 1, highlighting two key derived components: basic executors and executor composers with examples.

### 3.1 FINE-TUNING PROCESS AND IMPLEMENTATION DESIGN

CAEF offers a framework to enhance the arithmetic capabilities of LLMs. To implement Equation 1 for a specific arithmetic task, CAEF involves the following steps: 1) step 1: design a state machine and implement the representation $< s_i, c_i >$ for the arithmetic task, and 2) step 2: sample pairs of input and output to create a dataset, which is then used to fine-tune the LLM for one-step execution.

**Step 1**. Designing a state machine can draw from existing Turing machines or other relevant state machines for the task. To implement state $s_i$ and commands $c_i$ in the representation, we transform the structured representation into plain text following two main guidelines: 1) numbers are formatted in R2L order, separated by |, and 2) each command is expressed in the format $\{[CMD]\ action\}$. For example, for the addition task $45 + 67 =$ where the current step involves adding the tens digits

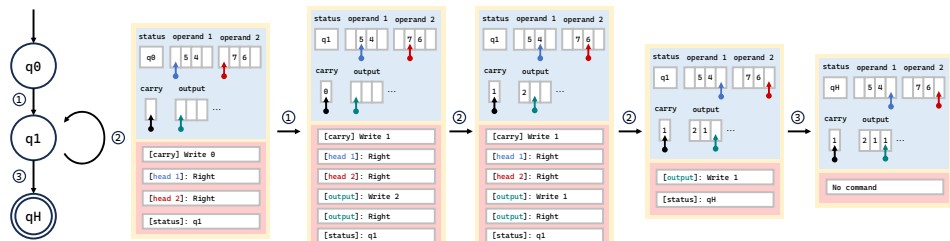

Figure 3: Execution process of $45 + 67$. The state diagram on the left abstracts the addition process. In step ②, a one-digit addition is performed, followed by updating the carry and output. The right side shows the actual sequence of state and command execution in the CAEF framework.

with a carry of 1 from the units place, the representation $< s_i, c_i >$ may include several pointers: two HEADs pointing to the digits, a carry $C$ for the carry value, and OUTPUT to record the results. All these pointers are moved using the RIGHT command. The representation is written as follows:

> $s_i$ : ADD, q1, | 5[HEAD1]| 4 | 7[HEAD2]| 6 [C]1 | 2[OUTPUT]
> $c_i$ : CMD: [C] 1, [OUTPUT] 1, [OUTPUT] RIGHT, [HEAD1] RIGHT, [HEAD2] RIGHT, q1

where q1 indicates the current status, and all pointers are presented in uppercase, enclosed in brackets with the pointed value on their right.

**Step 2**. To fine-tune the LLM, the dataset, including input and output representation pairs used for learning one-step execution. Continuing with the example, we create the representation $< s_{i+1}, c_{i+1} >$ for the output of the one-step execution based on the above $< s_i, c_i >$:

> $s_{i+1}$ : ADD, q1, | 5| 4[HEAD1] | 7| 6[HEAD2] [C]1 | 2| 1[OUTPUT]
> $c_{i+1}$ : CMD: [OUTPUT] 1, [OUTPUT], [C], qH

where qH is the halting status. The details of the dataset refer to Section 4.1.

One efficient way for LLM to learn for one-step execution is LoRA fine-tuning. Since we target to learn $+, -, \times, \div, >, <$, and $==$ arithmetic operators, implementing multiple LLM instances leads to significant memory overhead. To mitigate this, we use a single base LLM model with multiple LoRA adapters that serve as learned executable arithmetic operators. Switching LoRA adapters based on function calls generated by the LLM can efficiently perform various operations, optimizing memory usage while maintaining flexibility in handling different arithmetic computations.

To implement a specific computational task, CAEF introduces two types of executors (i.e., *basic executor* and the *executor composer*) to learn to execute under the proposed representation. The basic executor is designed to handle tasks with well-defined computational logic, while the executor composer acts as a higher-level controller that orchestrates the process by calling other basic executors. In the following, we introduce the two types of executors through illustrative examples.

## 3.2 BASIC EXECUTORS

We use *addition* to illustrate the design of a basic executor. A natural way to implement addition is to imitate the accumulator, performing the addition of two corresponding digits from the operands once at a time, along with the value stored in the carry register. This process calculates the result for the current digit and simultaneously updates the carry register for the next higher digit's computation.

Thus, the state and the command for addition are constructed as follows. The state should include the following components: 1) the two operands, 2) two pointers indicating the current digits being processed, 3) the carry register, and 4) the output generated so far. The command part should at least include: 1) the actions to write the carry and output, 2) the actions to move the pointers, and 3) state transition actions to control the start, transitions, and halting of the addition. Based on this instruction, CAEF constructs the state machine based on the text-based represented $< s_i, c_i >$. Figure 3 illustrates the computation process of CAEF for *addition*. The details of computations and dataset are listed in Appendix A.3 and Section 4.1, respectively. In this paper, we use similar procedures to design the operators for $>, <$ and $==$.

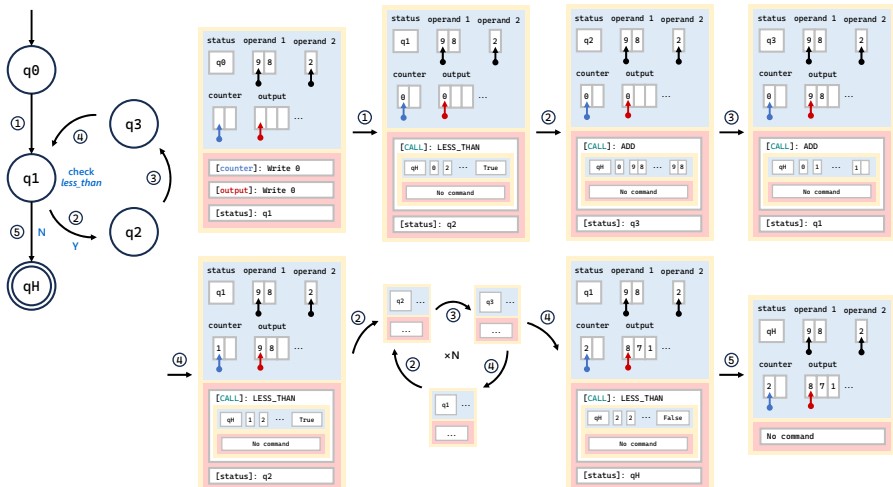

Figure 4: Execution process of $89 \times 2$. The state diagram on the left abstracts the multiplication process, where in state $q_1$, the less-than executor is performed. If true, the execution moves to state $q_2$; otherwise, it transitions to state $q_5$ and halts. Steps ③ and ④ execute the accumulation of the counter and output, respectively. The right side shows the actual execution in the CAEF framework.

### 3.3 EXECUTOR COMPOSERS

*Executor composer* designs to orchestrate the basic executors into intricate computational logic. Instead of performing computations directly, the executor composer "calls" other basic executors in a specific sequence to accomplish more complex tasks.

Multiplication is a typical example of the executor composer, which can be implemented by calling the $+$ and $<$ basic executors. CAEF uses two accumulators ($+$ involved) to implement $a \times b$. The first accumulator increments by $1$ with each loop iteration, while the second adds $a$ during each iteration. This process continues until the first accumulator reaches $b$ ($<$ involved), and then the value in the second accumulator represents the final result. LLM is fine-tuned to execute control flow and loops, by calling the $<$ executor and, based on its result, either halts or continues the loop. Figure 4 illustrates the computation process for $89 \times 2$ using our implementation. Since the executor composer decouples the computational logic into several executors, the fine-tuning process could be done separately for each executor, showing the ability of executor composition.

Besides *multiplication*, we also design *subtraction* (considering only non-negative results) and *division* (floor division) executor composers using similar methodologies. Specifically, we draw inspiration from how subtraction is handled in CPUs to construct the *subtraction* executor composer and the detailed implementation can be found in Appendix A.4.

## 4 EVALUATION

### 4.1 SETTING

**Models**. We utilize the LLaMA 3.1-8B pretrained model (non-instruct version) as the base model. During LoRA fine-tuning, all linear modules in the decoder layer are involved in training, with the hyperparameters fixed at $r = 8$, $\alpha = 16$, and a learning rate of $5 \times 10^{-5}$. The fine-tuning process is conducted in two stages. In the first stage, we introduce an exhaustive explanation in the prompt, detailing the computation goal of an executor, the required input/output format, and providing an example. This explanation is followed by the actual sample, as illustrated in Appendix A.1. In the second stage, we remove the long explanation from the prompt and present only the sample, expecting the model to predict the next state and the subsequent commands directly. We use batch sizes of 8 and 16 for the first and second stages, respectively. All experiments are conducted on a server equipped with six H800 GPUs. The code and the models are available[1].

---

[1]The implementation code is accessible at https://github.com/HNDRXwjrmY/CAEF, and the checkpoints are available at https://huggingface.co/HNDRXwjrmY/CAEF_llama3.1_8b

Table 1: Overall evaluation results across seven operators. ”LLaMA 3.1 (L)” refers to the LLaMA fine-tuned with LoRA, while ”LLaMA 3.1 (I)” refers to the LLaMA 3.1-8B-Instruct model.

| Operator | Model | 5-digits | 10-digits | 50-digits | 100-digits | 1~10-digits |
|---|---|---|---|---|---|---|
| Addition | CAEF | 100.0 | 99.6 | 99.9 | 98.6 | - |
| | LLaMA 3.1 (L) | $92.1_{-7.9}$ | $64.8_{-34.8}$ | $0.0_{-99.9}$ | $0.0_{-98.6}$ | - |
| | LLaMA 3.1 (I) | $93.5_{-6.5}$ | $35.0_{-64.6}$ | $0.0_{-99.9}$ | $0.0_{-98.6}$ | - |
| | GPT-4o | $98.4_{-1.6}$ | $94.0_{-5.6}$ | $65.0_{-34.9}$ | $43.0_{-55.6}$ | - |
| Subtraction | CAEF | 98.7 | 99.5 | 98.8 | 98.0 | - |
| | LLaMA 3.1 (L) | $82.8_{-15.9}$ | $61.0_{-38.5}$ | $0.0_{-98.8}$ | $0.0_{-98.0}$ | - |
| | LLaMA 3.1 (I) | $92.6_{-6.1}$ | $60.3_{-39.2}$ | $0.0_{-98.8}$ | $0.0_{-98.0}$ | - |
| | GPT-4o | $98.6_{-0.1}$ | $95.9_{-3.6}$ | $84.0_{-14.8}$ | $71.6_{-26.4}$ | - |
| Greater_than | CAEF | 99.2 | 99.0 | 99.2 | 97.2 | - |
| | LLaMA 3.1 (L) | $93.0_{-6.2}$ | $90.0_{-9.0}$ | $46.3_{-52.9}$ | $10.0_{-87.2}$ | - |
| | LLaMA 3.1 (I) | $99.3_{+0.1}$ | $97.7_{-1.3}$ | $72.1_{-27.1}$ | $70.0_{-27.2}$ | - |
| | GPT-4o | $99.8_{+0.6}$ | $99.6_{+0.6}$ | $99.0_{-0.2}$ | $93.2_{-4.0}$ | - |
| Less_than | CAEF | 99.7 | 99.3 | 99.6 | 98.0 | - |
| | LLaMA 3.1 (L) | $96.2_{-3.5}$ | $93.6_{-5.7}$ | $84.0_{-15.6}$ | $45.0_{-53.0}$ | - |
| | LLaMA 3.1 (I) | $93.9_{-5.8}$ | $86.3_{-13.0}$ | $74.6_{-25.0}$ | $67.4_{-30.6}$ | - |
| | GPT-4o | $99.9_{+0.2}$ | $100.0_{+0.7}$ | $99.3_{-0.3}$ | $89.2_{-8.8}$ | - |
| Equal | CAEF | 99.4 | 99.6 | 99.1 | 98.4 | - |
| | LLaMA 3.1 (L) | $57.5_{-41.9}$ | $66.2_{-33.4}$ | $59.2_{-39.9}$ | $54.0_{-44.4}$ | - |
| | LLaMA 3.1 (I) | $100.0_{+0.6}$ | $98.8_{-0.8}$ | $99.6_{+0.5}$ | $99.6_{+1.2}$ | - |
| | GPT-4o | $100.0_{+0.6}$ | $100.0_{+0.4}$ | $100.0_{+0.9}$ | $100.0_{+1.6}$ | - |
| Multiplication | CAEF | - | - | - | - | 99.3 |
| | LLaMA 3.1 (L) | - | - | - | - | $61.8_{-37.5}$ |
| | LLaMA 3.1 (I) | - | - | - | - | $61.4_{-37.9}$ |
| | GPT-4o | - | - | - | - | $97.7_{-1.6}$ |
| Division | CAEF | - | - | - | - | 99.3 |
| | LLaMA 3.1 (L) | - | - | - | - | $98.4_{-0.9}$ |
| | LLaMA 3.1 (I) | - | - | - | - | $96.5_{-2.8}$ |
| | GPT-4o | - | - | - | - | $99.1_{-0.2}$ |

**Baseline**. We compare our approach against three baselines on $+$, $-$, $\times$, $\div$, $==$, $>$, and $<$ operators. The first is a LLM fine-tuned with LoRA on LLaMA 3.1-8B (non-instruct version). Additionally, we include two unmodified LLMs, GPT-4o and LLaMA 3.1-8B Instruct, both of which directly generate the computational results based on the arithmetic expressions through a single model query. The prompts used for these models are in Appendix A.5.

**Dataset**. In CAEF, an operator requires an executor and an aligner, each supported by a specific LoRA adapter. To generate training datasets for these adapters, we implement a Turing machine prototype for each operator. For the executor, we generate random arithmetic expressions and run the Turing machine from its initial state until it halts, recording states and commands before and after each transition. This produces a sequence of states and commands, from which we sampled to train the executor. By generating multiple sequences through random initialization, an adequate training dataset for the executor can be created. It is notable that for arithmetic expressions with long operands, the sequences tend to be lengthy. Simple random sampling may lead to a dataset dominated by intermediate steps, potentially omitting samples from the first and final transitions. To address this, we ensure that the first and last steps are always included. Similarly, for the aligner, we generate two alignment processes: one aligning the original arithmetic expression with the executor's initial state and first command, while another aligning the executor's halt state with the final result of the original arithmetic expression.

For the test sets, we generate a dataset consisting of pure arithmetic expressions using predefined templates (refer to in Appendix A.2). Specifically, for $+$, $-$, $==$, $>$, and $<$ operations, we create test sets with two operands of equal length, consisting of 5, 10, 50, and 100 digits. For multiplication and division, to avoid excessively large values, we adjusted the data range based on the characteristics of these two operators. In multiplication of the form $a \times b = c$, we restricted $a$ to be a random number

Table 2: Accuracy of the executor and aligner across seven operators. The executor's accuracy refers to the probability of completing the entire computation correctly from the initial state to the final step, with each step being accurate. The aligner's accuracy is divided into two parts: the conversion from the original input to the executor's representation, denoted as aligner (I), and the conversion from the executor's final representation to the output, denoted as aligner (O).

| Operator | Component | 5-digits | 10-digits | 50-digits | 100-digits | 1∼10-digits |
|---|---|---|---|---|---|---|
| Addition | executor | 100.0 | 100.0 | 99.9 | 99.6 | - |
|  | aligner (I) | 100.0 | 99.7 | 100.0 | 99.6 | - |
|  | aligner (O) | 100.0 | 99.9 | 100.0 | 99.4 | - |
| Subtraction | executor | 100.0 | 100.0 | 99.6 | 99.2 | - |
|  | aligner (I) | 98.8 | 99.7 | 99.5 | 99.6 | - |
|  | aligner (O) | 99.9 | 99.7 | 99.7 | 99.2 | - |
| Greater_than | executor | 100.0 | 100.0 | 99.8 | 99.6 | - |
|  | aligner (I) | 99.2 | 99.1 | 99.4 | 98.6 | - |
|  | aligner (O) | 100.0 | 99.9 | 100.0 | 99.2 | - |
| Less_than | executor | 100.0 | 100.0 | 100.0 | 100.0 | - |
|  | aligner (I) | 99.8 | 99.3 | 99.7 | 98.4 | - |
|  | aligner (O) | 99.9 | 100.0 | 99.8 | 99.6 | - |
| Equal | executor | 100.0 | 100.0 | 99.8 | 99.4 | - |
|  | aligner (I) | 99.4 | 99.6 | 99.6 | 98.8 | - |
|  | aligner (O) | 100.0 | 100.0 | 99.8 | 99.8 | - |
| Multiplication | executor | - | - | - | - | 99.5 |
|  | aligner (I) | - | - | - | - | 99.8 |
|  | aligner (O) | - | - | - | - | 100.0 |
| Division | executor | - | - | - | - | 99.4 |
|  | aligner (I) | - | - | - | - | 100.0 |
|  | aligner (O) | - | - | - | - | 99.9 |

with 1-10 digits, and $b$ to fall within the value range $[1, 15]$. In division of the form $a \div b = c$, we constrained $c$ to be within $[1, 15]$ and $b$ to be a random number with 1-10 digits.

**Metrics**. We employ accuracy as the evaluation metric. Each arithmetic problem is computed once, and the result is compared with the ground truth using the Exact Match criterion.

## 4.2 MAIN RESULTS

Table 1 presents the evaluation results of our method and baseline models across the seven operators. Compared to the baselines, the proposed approach performs stably on all operators with high accuracy. Specifically for tasks with long numbers, such as 100-digit addition, LLM with CAEF effectively learns the computational logic to execute the addition process.

To further explore the actual performance of the executor and aligner during the computation process, we separately evaluate their accuracy on the same dataset. As the results shown in Table 2, we observe that even though the executor must generate numerous intermediate steps in an iterative manner, while the aligner only performs two conversion steps, the executor still outperforms the aligner overall. The executor achieves over 99% accuracy in all experimental settings, indicating that it has effectively learned the arithmetic logic. When provided with the correct initial state and command, it functions correctly in the vast majority of cases. On the other hand, the aligner shows lower accuracy when converting the original input compared to converting the executor's output in most cases, suggesting that the bottleneck in the overall computation process lies in the reversal of operands, rather than in the computation itself. Due to the page limit, more detailed analysis are presented in Appendix A.6, the analysis of computational complexity in CAEF is detailed in A.7, and an extended experiment exploring the merging of the aligner and executor for individual operators is presented in A.8.

## 5 RELATED WORK

**LLMs in Mathematical Contexts**. Prior research has focused on enhancing LLM performance in mathematical tasks, often relying on external tools for calculations and primarily addressing math word problems rather than pure arithmetic. A common external tool is a calculator, as exemplified by Schick et al. (2024), which introduces a self-supervised method where the model learns when to call external tools via API access. Similar strategies can be found in Gou et al. (2023) and He-Yueya et al. (2023), and it was employed in even earlier work (Andor et al., 2019). Another tool is a programming language interpreter, such as Python, where the model generates code and an external interpreter executes it to obtain the result. A representative example is Lu et al. (2024), which treats the LLM as a planner that generates code and submits it to an external Python executor to handle math problems in tabular contexts. Wang et al. (2023) employs supervised fine-tuning to improve code-based problem-solving, while Zhou et al. (2023) proposes a zero-shot prompting method to enable code-driven self-verification, thereby improving mathematical performance.

**LLMs in Arithmetic Scenarios**. Another series of work focuses solely on arithmetic, which we consider directly related to our research. The common characteristic of these studies is their effort to teach LLMs computational logic and improve calculation accuracy through step-by-step processes. Among these works, Nye et al. (2021) is an early and far-reaching study, predating the popular Chain-of-Thought (CoT) approach. It introduces a similar idea in the arithmetic domain, where the language model outputs intermediate steps to a buffer called a "scratchpad," significantly improving performance in integer addition. Hu et al. (2024) observes that transformers tend to approach arithmetic problems using "case-based reasoning" and proposes a Rule-Following Fine-Tuning technique that guides the model to execute calculations step by step. Zhou et al. (2024) combines four techniques (i.e., FIRE position encodings, Randomized position encodings, Reversed format (R2L format), and Index hints) to develop a new model that achieves a $2.5\times$ improvement in length generalization for two-integer addition.

## 6 LIMITATIONS

**Prone to errors with repeated digit patterns**. Both the executor and the aligner tend to generate incorrect steps when encountering patterns of repeated digits, such as sequences like "999..." where a single digit repeats, or "456456..." where multiple digits repeat. These errors typically manifest as extra or missing repetitions of the pattern. While this issue might be mitigated by intentionally generating more such expressions to increase the representation of similar samples in the training set, we believe the root cause lies in limitations inherent to generative LLMs.

**Efficiency Issue**. In our method, completing a single computation requires generating the full sequence of intermediate steps, which essentially means repeatedly calling the *model.generate()* function. For computations involving hundreds of steps, this process can be extremely time-consuming. One potential solution lies in optimizing the use of the KV cache. In our approach, the input to the LLM at two consecutive steps is highly similar. However, since different parts of the input shift position, the KV cache from the previous step cannot be effectively reused. The KV cache functions like a ROM. If we could transform it into a RAM-like structure that supports simple editing operations, such as swapping adjacent tokens while maintaining the correct tokens and positional embeddings, this could significantly improve computational efficiency.

**Implementation of the Turing machine prototype**. When generating the training set for the executor, CAEF wants to ensure the correctness of the samples and enable the executor to learn key computational steps, such as carrying over or exiting loops. A practical approach is to construct a Turing machine prototype corresponding to the target operator and record its execution process. While there are many existing Turing machine designs, the implementation process may take some human-involved effort. A future work could design a generation process to translate existing Turing machines into CAEF required Turing machine prototypes.

**Inability to Solve Math Word Problems.** Currently, our method requires manual selection of the active LoRA adapter, rather than enabling the model to autonomously determine the appropriate adapter. This limitation hinders the direct application of our approach to solving math word problems. However, our method can be regarded as a modular component. Several studies have explored integrating Mixture of Experts (MoE) and LoRA techniques to facilitate the automatic

selection and switching of active LoRA adapters based on the input (Wu et al., 2023a; Zadouri et al., 2023; Huang et al., 2023; Xu et al., 2024). These studies are orthogonal to our approach, and we posit that combining these techniques with our method could enable effective application to math word problems. For example, leveraging the CAEF plug-in, a large language model (LLM) could dynamically switch to CAEF to handle arithmetic computations as part of the reasoning process in solving such problems.

## 7 CONCLUSION

This paper proposes a framework that enables LLMs to learn to execute step-by-step arithmetic computational logic by imitating the behavior of a Turing machine. This approach significantly enhances LLMs' computational capability without relying on any external tools. Moreover, the framework is highly scalable, allowing the construction of complex executors by composing learned basic executors, reducing the difficulty of learning the complex logic. We hope that our work provides a new perspective for enabling LLMs to learn rule-based computation.

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

# A APPENDIX

## A.1 EXAMPLE OF SAMPLES IN TRAINING SET OF EXECUTOR AND ALIGNER

### A.1.1 ADDITION

Addition executor:

---

*Input*:
The following is a state paired with a command to be executed of a Turing Machine that performs addition.

The state includes the current operator, the current state of the machine, the current tape contents, and the current head positions.
- There are three states in the machine: q0, q1, and qH. The machine starts in state q0 and halts when it reaches state qH. q1 is the state where the machine does the addition and calculates the carry out.
- The head positions are represented by [HEAD1] and [HEAD2], which indicate the positions of the heads on the two operands.
- The carry out is represented by [C].
- The output position is represented by [OUTPUT].

The command includes a series of actions to be executed by the machine and they are separated by commas.
- [OUTPUT] <number>: Write the number to the output position.
- [OUTPUT] <direction>: Move the output head to the direction.
- [C] <number>: Write the number to the carry out register.
- [HEAD1] <direction>: Move the head on the first operand to the direction.
- [HEAD2] <direction>: Move the head on the second operand to the direction.
- <state>: Move the machine to the state.

The machine performs addition by reading the digits from the two operands and writing the sum to the output tape.

Based the current state and the command, predict the next state of the machine and next command to be executed.

ADD, q0, [HEAD1] |5|4[HEAD2] |7|6 [C] [OUTPUT]
CMD: [C] 0, [HEAD1] RIGHT, [HEAD2] RIGHT, q1

*Output*:
ADD, q1, [HEAD1]|5|4 [HEAD2]|7|6 [C]0 [OUTPUT]
CMD: [C] 1, [OUTPUT] 2, [OUTPUT] RIGHT, [HEAD1] RIGHT, [HEAD2] RIGHT, q1

---

Addition aligner:

---

*Input*:
The following is an input to a Turing Machine or an output of a Turing Machine.
The task is doing an adaptation:
- If it is an input, adapt the original input to the format that the Turing Machine can understand.
- If it is an output, adapt the original output to the format that represents the final result.

Input example:
"'
- input:
1504+2379=

---

- output:
ADD, q0, [HEAD1] |4|0|5|1[HEAD2] |9|7|3|2 [C] [OUTPUT]
CMD: [C] 0, [HEAD1] RIGHT, [HEAD2] RIGHT, q1
```

Output example:
```
- input:
ADD, qH, |7|6|3|4[HEAD1] |4|3|2|1[HEAD2] [C]0 |1|0|6|5
No command to execute. Halt state.
- output:
4367+1234=5601
```

There are two lines that represent the Turing Machine:
- The first line is the current state of the machine.
- The second line is the command to be executed.
And this format is fit to both input and output as the examples shown above.

For the current state (the first line):
- There are at least 2 states in the machine: q0 and qH. The machine starts in state q0 and halts when it reaches state qH.
- The head positions are represented by [HEAD1] and [HEAD2], which followed by two operands.
- [C] represents the carry out register and [OUTPUT] represents the output position. And these two are empty at the beginning.

The command (the second line) includes a series of actions to be executed by the machine and they are separated by commas.
- [HEAD] <direction>: Move the head to the direction.
- [C] <number>: Write the number to the carry out register.
- <state>: Move the machine to the state.

Note that the number is represented in reverse order in machine, which is beneficial to the machine to perform the subtraction operation.
Based on the input, determine it is an input or an output, and adapt it to the format correspondingly.

45+67=

*Output*:
ADD, q0, [HEAD1] |5|4[HEAD2] |7|6 [C] [OUTPUT]
CMD: [C] 0, [HEAD1] RIGHT, [HEAD2] RIGHT, q1

### A.1.2 SUBTRACTION

Subtraction executor:

*Input*:
The following is a input to be executed of a Turing Machine that performs subtraction.

To solve a subtraction problem by the machine, the machine is required to provide the initial state and command for other basic machines, including addition, reflection and left mask.

For example, for 47819 - 12345 = 35474, the machine will perform the following steps:
- step 1: call reflection, 99999 - 12345 = 87654
- step 2: call addition, 47819 + 87654 = 135473
- step 3: call addition, 135473 + 1 = 135474

- step 4: call left mask, left_mask(135474) = 35474

The input may includes four lines or the original subtraction problem.
When it is original problem, generate the initial subtraction state, command and prepare the initial state and the first command of the first called machine.
When it includes four lines, it means the previous state, command and the result of the called machine. In detail:
- The first line is the current state of the machine.
- The second line is the command to be executed.
- The third line and the fourth line are halt state of another machine which is called by the subtraction machine at previous step.

For the current state (the first line):
- There are five states in the machine: q0, q1, q2, q3 and qH. The machine starts in state q0 and halts when it reaches state qH.
- The head positions are represented by [HEAD1] and [HEAD2], which followed by two operands.

The command (the second line) includes a series of actions to be executed by the machine and they are separated by commas.
- [CALL] <operation>: Call another machine to perform the operation.
- <state>: Move the machine to the state.

When the commands include [CALL], another extra two lines are needed to specify the initial state and the first command of the machine to be called.
As for initial state, it should include the operation, q0 state, operands and the head positions.
As for the first command:
- [OUTPUT] <number>: Write the number to the output position.
- [OUTPUT] <direction>: Move the output head to the direction.
- [HEAD1] <direction>: Move the head on the first operand to the direction.
- [HEAD2] <direction>: Move the head on the second operand to the direction.
- <state>: Move the machine to the state.

The machine performs subtraction by reading the digits from the two operands and calling other machines to complete the subtraction operation.

Based on the current input, predict the output which includes next state, next command and the initial state and the first command of the machine to be called.

SUB, q0, [HEAD1]|7|4 [HEAD2]|2|1
CMD q1

*Output*:
SUB, q1, [HEAD1]|7|4 [HEAD2]|2|1
CMD [CALL] REFLECTION, q2
REFLECTION, q0, [HEAD1] |9|9[HEAD2] |2|1 [OUTPUT]
CMD [HEAD1] RIGHT, [HEAD2] RIGHT, q1

Subtraction aligner:

*Input*:
The following is an input to a Turing Machine or an output of a Turing Machine.

The task is doing an adaptation:
- If it is an input, adapt the original input to the format that the Turing Machine can understand.

- If it is an output, adapt the original output to the format that represents the final result.

Input example:
```
- input:
4531-1504=
- output:
SUB, q0, [HEAD1]|1|3|5|4 [HEAD2]|4|0|5|1
CMD q1
```

Output example:
```
- input:
SUB, qH, [HEAD1]|1|3|5|4 [HEAD2]|4|0|5|1 |7|2|0|3
No command to execute. Halt state.
- output:
4531-1504=3027
```

There are two lines that represent the Turing Machine:
- The first line is the current state of the machine.
- The second line is the command to be executed.
And this format is fit to both input and output as the examples shown above.

For the current state (the first line):
- There are at least 2 states in the machine: q0 and qH. The machine starts in state q0 and halts when it reaches state qH.
- The head positions are represented by [HEAD1] and [HEAD2], which followed by two operands.

The command (the second line) includes a series of actions to be executed by the machine and they are separated by commas.
- [HEAD] <direction>: Move the head to the direction.
- [OUTPUT] <number>: Write the number to the output position.
- <state>: Move the machine to the state.

Note that the number is represented in reverse order in machine, which is beneficial to the machine to perform the subtraction operation.

Based on the input, determine it is an input or an output, and adapt it to the format correspondingly.

46-28=

*Output*:
SUB, q0, [HEAD1]|6|4 [HEAD2]|8|2
CMD q1

### A.1.3 MULTIPLICATION

Multiplication executor:

*Input*:
The following is a input to be executed of a Turing Machine that performs multiplication.

To solve a multiplication problem by the machine, the machine is required to provide the initial state and command for other basic machines, including addition and less_than machines.

For example, for 4513 * 3 = 13539, the machine will perform the following algorithm:
- step 1: cnt = 1, sum = 4513(oprand1)
- step 2: call less_than, determine whether cnt <3(oprand2), if yes, go to step 3, otherwise, go to step 5
- step 3: call addition, sum = sum + 4513(oprand1)
- step 4: call addition, cnt = cnt + 1, go to step 2
- step 5: current machine halts

The input includes at least two lines and may have two more lines.
- The first line is the current state of the machine.
- The second line is the command to be executed.
When there are two more lines:
- The third line and the fourth line are halt state of another machine which is called by the multiplication machine at previous step.

For the current state (the first line):
- There are five states in the machine: q0, q1, q2, q3 and qH. The machine starts in state q0 and halts when it reaches state qH. q1, q2 and q3 are used to perform the loop structure.
- The head positions are represented by [HEAD1] and [HEAD2], which followed by two operands.

The command (the second line) includes a series of actions to be executed by the machine and they are separated by commas.
- [OUTPUT] <number>: Write the number to the output position.
- [COUNT] <number>: Write the number to the count register.
- [CALL] <operation>: Call another machine to perform the operation.
- <state>: Move the machine to the state.

When the commands include [CALL], another extra two lines are needed to specify the initial state and the first command of the machine to be called.
As for initial state, it should include the operation, q0 state, operands and the head positions.
As for the first command:
- [OUTPUT] <number>: Write the number to the output position.
- [OUTPUT] <direction>: Move the output head to the direction.
- [HEAD1] <direction>: Move the head on the first operand to the direction.
- [HEAD2] <direction>: Move the head on the second operand to the direction.
- <state>: Move the machine to the state.

The machine performs multiplication by reading the digits from the two operands and calling other machines to complete the multiplication operation.

Based on the current input, predict the output which includes next state, next command and the initial state and the first command of the machine to be called.

MUL, q3, [HEAD1]|3|8|6 [HEAD2]|8|6 [COUNT]|5|4 [OUTPUT]|8|1|4|1|3
CMD [CALL] ADD, q1
ADD, qH, |5|4[HEAD1] |1[HEAD2] [C]0 |6|4
No command to execute. Halt state.

*Output*:
MUL, q1, [HEAD1]|3|8|6 [HEAD2]|8|6 [COUNT]|6|4 [OUTPUT]|8|1|4|1|3
CMD [CALL] LESS_THAN, q2

LESS_THAN, q0, [HEAD1]|6|4[HEAD2] |8|6 [OUTPUT]
CMD [HEAD1] RIGHT, [HEAD2] RIGHT, [OUTPUT] False, q1

Multiplication aligner:

*Input*:
The following is an input to a Turing Machine or an output of a Turing Machine.

The task is doing an adaptation:
- If it is an input, adapt the original input to the format that the Turing Machine can understand.
- If it is an output, adapt the original output to the format that represents the final result.

Input example:
```
- input:
44814*5=
- output:
MUL, q0, [HEAD1]|4|1|8|4|4 [HEAD2]|5 [COUNT] [OUTPUT]
CMD [COUNT] 1, [OUTPUT]|4|1|8|4|4, q1
```

Output example:
```
- input:
MUL, qH, [HEAD1]|4|1|8|4|4 [HEAD2]|5 [COUNT]|5 |0|7|0|4|2|2
No command to execute. Halt state.
- output:
44814*5=224070
```

There are two lines that represent the Turing Machine:
- The first line is the current state of the machine.
- The second line is the command to be executed.
And this format is fit to both input and output as the examples shown above.

For the current state (the first line):
- There are at least 2 states in the machine: q0 and qH. The machine starts in state q0 and halts when it reaches state qH.
- The head positions are represented by [HEAD1] and [HEAD2], which followed by two operands.

The command (the second line) includes a series of actions to be executed by the machine and they are separated by commas.
- [HEAD] <direction>: Move the head to the direction.
- [OUTPUT] <number>: Write the number to the output position.
- [COUNT] <number>: Write the number to the count register.
- <state>: Move the machine to the state.

Based on the input, determine it is an input or an output, and adapt it to the format correspondingly.

652202674*9560505=

*Output*:
MUL, q0, [HEAD1]|4|7|6|2|0|2|2|5|6 [HEAD2]|5|0|5|0|6|5|9 [COUNT] [OUTPUT]

CMD [COUNT] 0, [OUTPUT] 0, q1

### A.1.4 DIVISION

Division executor:

*Input*:
The following is a input to be executed of a Turing Machine that performs division.

To solve a division problem by the machine, the machine is required to provide the initial state and command for other basic machines, including addition and greater_than machines.

For example, for 4513 // 1504 = 3, the machine will perform the following algorithm:
- step 1: output = 0, cnt = 1504(oprand2)
- step 2: call greater_than, determine whether cnt >4513(oprand1), if yes, go to step 5, otherwise, go to step 3
- step 3: call addition, output = output + 1
- step 4: call addition, cnt = cnt + 1504, go to step 2
- step 5: current machine halts, output is the result

The input includes at least two lines and may have two more lines.
- The first line is the current state of the machine.
- The second line is the command to be executed.
When there are two more lines:
- The third line and the fourth line are halt state of another machine which is called by the division machine at previous step.

For the current state (the first line):
- There are five states in the machine: q0, q1, q2, q3 and qH. The machine starts in state q0 and halts when it reaches state qH. q1, q2 and q3 are used to perform the loop structure.
- The head positions are represented by [HEAD1] and [HEAD2], which followed by two operands.

The command (the second line) includes a series of actions to be executed by the machine and they are separated by commas.
- [OUTPUT] <number>: Write the number to the output position.
- [COUNT] <number>: Write the number to the count register.
- [CALL] <operation>: Call another machine to perform the operation.
- <state>: Move the machine to the state.

When the commands include [CALL], another extra two lines are needed to specify the initial state and the first command of the machine to be called.
As for initial state, it should include the operation, q0 state, operands and the head positions.
As for the first command:
- [OUTPUT] <number>: Write the number to the output position.
- [OUTPUT] <direction>: Move the output head to the direction.
- [HEAD1] <direction>: Move the head on the first operand to the direction.
- [HEAD2] <direction>: Move the head on the second operand to the direction.
- <state>: Move the machine to the state.

The machine performs division by reading the digits from the two operands and calling other machines to complete the division operation.

Based on the current input, predict the output which includes next state, next command and

the initial state and the first command of the machine to be called.

DIV, q1, [HEAD1]|0|5|6 [HEAD2]|8|3|2 [COUNT]|6|7|4 [OUTPUT]|1
CMD [CALL] GREATER_THAN, q2
GREATER_THAN, qH, |6|7|4[HEAD1] |0|5|6[HEAD2] False
No command to execute. Halt state.

*Output*:
DIV, q2, [HEAD1]|0|5|6 [HEAD2]|8|3|2 [COUNT]|6|7|4 [OUTPUT]|1
CMD [CALL] ADD, q3
ADD, q0, [HEAD1] |1[HEAD2] |1 [C] [OUTPUT]
CMD: [C] 0, [HEAD1] RIGHT, [HEAD2] RIGHT, q1

Division aligner:

*Input*:
The following is an input to a Turing Machine or an output of a Turing Machine.

The task is doing an adaptation:
- If it is an input, adapt the original input to the format that the Turing Machine can understand.
- If it is an output, adapt the original output to the format that represents the final result.

Input example:
```
- input:
4531//1504=
- output:
DIV, q0, [HEAD1]|3|1|5|4 [HEAD2]|4|0|5|1 [COUNT] [OUTPUT]
CMD [COUNT]|4|0|5|1, [OUTPUT] 0, q1
```

Output example:
```
- input:
DIV, qH, [HEAD1]|3|1|5|4 [HEAD2]|4|0|5|1 [COUNT]|6|1|0|6 |3
No command to execute. Halt state.
- output:
4531//1504=3
```

There are two lines that represent the Turing Machine:
- The first line is the current state of the machine.
- The second line is the command to be executed.
And this format is fit to both input and output as the examples shown above.

For the current state (the first line):
- There are at least 2 states in the machine: q0 and qH. The machine starts in state q0 and halts when it reaches state qH.
- The head positions are represented by [HEAD1] and [HEAD2], which followed by two operands.

The command (the second line) includes a series of actions to be executed by the machine and they are separated by commas.
- [HEAD] <direction>: Move the head to the direction.
- [OUTPUT] <number>: Write the number to the output position.

- [COUNT] <number>: Write the number to the count register.
- <state>: Move the machine to the state.

Based on the input, determine it is an input or an output, and adapt it to the format correspondingly.

8634010848//613431149=

*Output*:
DIV, q0, [HEAD1]|8|4|8|0|1|0|4|3|6|8 [HEAD2]|9|4|1|1|3|4|3|1|6 [COUNT] [OUTPUT]
CMD [COUNT]|9|4|1|1|3|4|3|1|6, [OUTPUT] 0, q1

### A.1.5 GREATER_THAN

Greater_than executor:

*Input*:
The following is a state paired with a command to be executed of a Turing Machine that determines whether the first operand is greater than the second operand.

The state includes the current operator, the current state of the machine, the current tape contents, and the current head positions.
- There are three states in the machine: q0, q1, and qH. The machine starts in state q0 and halts when it reaches state qH. q1 is the state where the machine does the comparison.
- The head positions are represented by [HEAD1] and [HEAD2], which indicate the positions of the heads on the two operands.
- The output position is represented by [OUTPUT].

The command includes a series of actions to be executed by the machine and they are separated by commas.
- [OUTPUT] <number>: Write the number to the output position.
- [OUTPUT] <direction>: Move the output head to the direction.
- [HEAD1] <direction>: Move the head on the first operand to the direction.
- [HEAD2] <direction>: Move the head on the second operand to the direction.
- <state>: Move the machine to the state.

The machine performs comparison by reading the digits from the two operands and writing the result to the output tape.

Based on the current state and the command, predict the next state of the machine and next command to be executed.

GREATER_THAN, q1, |1|7|6|7|0[HEAD1]|5|1|3|1 |5|6|4|1|7[HEAD2]|8|1|4|7|4|8|8|3|2|7 [OUTPUT]False
CMD [HEAD1] RIGHT, [HEAD2] RIGHT, [OUTPUT] False, q1

*Output*:
GREATER_THAN, q1, |1|7|6|7|0|5[HEAD1]|1|3|1 |5|6|4|1|7|8[HEAD2]|1|4|7|4|8|8|3|2|7 [OUTPUT]False
CMD [HEAD1] RIGHT, [HEAD2] RIGHT, q1

Greater_than aligner:

*Input*:
The following is an input to a Turing Machine or an output of a Turing Machine.

The task is doing an adaptation:
- If it is an input, adapt the original input to the format that the Turing Machine can understand.
- If it is an output, adapt the original output to the format that represents the final result.

Input example:
```
- input:
45131>15040=
- output:
GREATER_THAN, q0, [HEAD1] |1|3|1|5|4[HEAD2] |0|4|0|5|1 [OUTPUT]
CMD [HEAD1] RIGHT, [HEAD2] RIGHT, [OUTPUT] False, q1
```

Output example:
```
- input:
GREATER_THAN, qH, |1|3|1|5|4[HEAD1] |0|4|0|5|1[HEAD2] True
No command to execute. Halt state.
- output:
45131>15040=True
```

There are two lines that represent the Turing Machine:
- The first line is the current state of the machine.
- The second line is the command to be executed.
And this format is fit to both input and output as the examples shown above.

For the current state (the first line):
- There are at least 2 states in the machine: q0 and qH. The machine starts in state q0 and halts when it reaches state qH.
- The head positions are represented by [HEAD1] and [HEAD2], which followed by two operands.

The command (the second line) includes a series of actions to be executed by the machine and they are separated by commas.
- [HEAD] <direction>: Move the head to the direction.
- [OUTPUT] <direction>: Move the output head to the direction.
- [OUTPUT] <result>: Write the result to the output position.
- <state>: Move the machine to the state.

Based on the input, determine it is an input or an output, and adapt it to the format correspondingly.

46989>82541=

*Output*:
GREATER_THAN, q0, [HEAD1] |9|8|9|6|4[HEAD2] |1|4|5|2|8 [OUTPUT]
CMD [HEAD1] RIGHT, [HEAD2] RIGHT, [OUTPUT] False, q1

### A.1.6 LESS_THAN

Less_than executor:

*Input*:
The following is a state paired with a command to be executed of a Turing Machine that determines whether the first operand is less than the second operand.

The state includes the current operator, the current state of the machine, the current tape contents, and the current head positions.
- There are three states in the machine: q0, q1, and qH. The machine starts in state q0 and halts when it reaches state qH. q1 is the state where the machine does the comparison.
- The head positions are represented by [HEAD1] and [HEAD2], which indicate the positions of the heads on the two operands.
- The output position is represented by [OUTPUT].

The command includes a series of actions to be executed by the machine and they are separated by commas.
- [OUTPUT] <number>: Write the number to the output position.
- [OUTPUT] <direction>: Move the output head to the direction.
- [HEAD1] <direction>: Move the head on the first operand to the direction.
- [HEAD2] <direction>: Move the head on the second operand to the direction.
- <state>: Move the machine to the state.

The machine performs comparison by reading the digits from the two operands and writing the result to the output tape.

Based on the current state and the command, predict the next state of the machine and next command to be executed.

LESS_THAN, q1, |4|1|0[HEAD1]|2 |0|6|1[HEAD2]|2|7|6 [OUTPUT]True
CMD [HEAD1] RIGHT, [HEAD2] RIGHT, q1

*Output*:
LESS_THAN, q1, |4|1|0|2[HEAD1] |0|6|1|2[HEAD2]|7|6 [OUTPUT]True
CMD [OUTPUT] True, [OUTPUT], qH

Less_than aligner:

*Input*:
The following is an input to a Turing Machine or an output of a Turing Machine.

The task is doing an adaptation:
- If it is an input, adapt the original input to the format that the Turing Machine can understand.
- If it is an output, adapt the original output to the format that represents the final result.

Input example:
```
- input:
47182<83911=
- output:
LESS_THAN, q0, [HEAD1] |2|8|1|7|4[HEAD2] |1|1|9|3|8 [OUTPUT]
CMD [HEAD1] RIGHT, [HEAD2] RIGHT, [OUTPUT] False, q1
```

Output example:
```
- input:
LESS_THAN, qH, |2|8|1|7|4[HEAD1] |1|1|9|3|8[HEAD2] True

No command to execute. Halt state.
- output:
47182<83911=True
```

There are two lines that represent the Turing Machine:
- The first line is the current state of the machine.
- The second line is the command to be executed.
And this format is fit to both input and output as the examples shown above.

For the current state (the first line):
- There are at least 2 states in the machine: q0 and qH. The machine starts in state q0 and halts when it reaches state qH.
- The head positions are represented by [HEAD1] and [HEAD2], which followed by two operands.

The command (the second line) includes a series of actions to be executed by the machine and they are separated by commas.
- [HEAD] <direction>: Move the head to the direction.
- [OUTPUT] <direction>: Move the output head to the direction.
- [OUTPUT] <result>: Write the result to the output position.
- <state>: Move the machine to the state.

Based on the input, determine it is an input or an output, and adapt it to the format correspondingly.

LESS_THAN, qH, |1|5|9|4|4|6[HEAD1]|6|2|1|3|5|8|0|9|8 |3|7|2|6|4|2[HEAD2] False
No command to execute. Halt state.

*Output*:
890853126644951<246273=False

### A.1.7 EQUAL

Equal executor:

*Input*:
The following is a state paired with a command to be executed of a Turing Machine that performs equality comparison.

The state includes the current operator, the current state of the machine, the current tape contents, and the current head positions.
- There are three states in the machine: q0, q1, and qH. The machine starts in state q0 and halts when it reaches state qH. q1 is the state where the machine does the equality comparison.
- The head positions are represented by [HEAD1] and [HEAD2], which indicate the positions of the heads on the two operands.
- The output position is represented by [OUTPUT].

The command includes a series of actions to be executed by the machine and they are separated by commas.
- [OUTPUT] <number>: Write the number to the output position.
- [OUTPUT] <direction>: Move the output head to the direction.
- [HEAD1] <direction>: Move the head on the first operand to the direction.
- [HEAD2] <direction>: Move the head on the second operand to the direction.
- <state>: Move the machine to the state.

The machine performs equality comparison by reading the digits from the two operands and writing the result to the output tape.

Based on the current state and the command, predict the next state of the machine and next command to be executed.

EQUAL, q1, |0|5[HEAD1]|9 |0|5[HEAD2]|9 [OUTPUT]True
CMD [HEAD1] RIGHT, [HEAD2] RIGHT, q1

*Output*:
EQUAL, q1, |0|5|9[HEAD1] |0|5|9[HEAD2] [OUTPUT]True
CMD [OUTPUT], qH

Equal aligner:

*Input*:
The following is an input to a Turing Machine or an output of a Turing Machine.

The task is doing an adaptation:
- If it is an input, adapt the original input to the format that the Turing Machine can understand.
- If it is an output, adapt the original output to the format that represents the final result.

Input example:
'''
- input:
45263==45263=
- output:
EQUAL, q0, [HEAD1] |3|6|2|5|4[HEAD2] |3|6|2|5|4 [OUTPUT]
CMD [HEAD1] RIGHT, [HEAD2] RIGHT, [OUTPUT] True, q1
'''

Output example:
'''
- input:
EQUAL, qH, |3|6|2|5|4[HEAD1] |3|6|2|5|4[HEAD2] True
No command to execute. Halt state.
- output:
45263==45263=True
'''

There are two lines that represent the Turing Machine:
- The first line is the current state of the machine.
- The second line is the command to be executed.
And this format is fit to both input and output as the examples shown above.

For the current state (the first line):
- There are at least 2 states in the machine: q0 and qH. The machine starts in state q0 and halts when it reaches state qH.
- The head positions are represented by [HEAD1] and [HEAD2], which followed by two operands.

The command (the second line) includes a series of actions to be executed by the machine and they are separated by commas.
- [HEAD] <direction>: Move the head to the direction.
- [OUTPUT] <direction>: Move the output head to the direction.

- [OUTPUT] <result>: Write the result to the output position.
- <state>: Move the machine to the state.

Note that the number is represented in reverse order in machine, which is beneficial to the machine to perform the subtraction operation.

Based on the input, determine it is an input or an output, and adapt it to the format correspondingly.

EQUAL, qH, |6|5|6|8|8|9|7|1|6|7|7|1|2[HEAD1] |6|5|6|8|8|9|7|1|6|7|7|1|2[HEAD2] True
No command to execute. Halt state.

*Output*:
2177617988656==2177617988656=True

## A.2 ARITHMETIC EXPRESSION TEMPLATE

Templates in Table 3 are used for generate arithmetic expressions in our experiment.

Table 3: Templates used for generating arithmetic expressions in training set and test set.

| Operator | Template |
|---|---|
| Addition | {op1}+{op2}= |
| Subtraction | {op1}-{op2}= |
| Multiplication | {op1}*{op2}= |
| Division | {op1}//{op2}= |
| Greater | {op1}>{op2}= |
| Less | {op1}<{op2}= |
| Equal | {op1}=={op2}= |

## A.3 FULL COMPUTATION PROCESS OF THE EXAMPLES

The followings are the full computation process of the examples in 3.2 and 3.3.

### A.3.1 ADDITION

*Step 1* (aligner):
45+67=

*Step 2* (executor):
$state_0$: ADD, q0, [HEAD1] |5|4[HEAD2] |7|6 [C] [OUTPUT]
$command_0$: CMD: [C] 0, [HEAD1] RIGHT, [HEAD2] RIGHT, q1

*Step 3* (executor):
$state_1$: ADD, q1, [HEAD1]|5|4 [HEAD2]|7|6 [C]0 [OUTPUT]
$command_1$: CMD: [C] 1, [OUTPUT] 2, [OUTPUT] RIGHT, [HEAD1] RIGHT, [HEAD2] RIGHT, q1

*Step 4* (executor):
$state_2$: ADD, q1, |5[HEAD1]|4 |7[HEAD2]|6 [C]1 |2[OUTPUT]
$command_2$: CMD: [C] 1, [OUTPUT] 1, [OUTPUT] RIGHT, [HEAD1] RIGHT, [HEAD2] RIGHT, q1

*Step 5* (executor):
$state_3$: ADD, q1, |5|4[HEAD1] |7|6[HEAD2] [C]1 |2|1[OUTPUT]
$command_3$: CMD: [OUTPUT] 1, [OUTPUT], [C], qH

*Step 6* (executor):
$state_4$: ADD, qH, |5|4[HEAD1] |7|6[HEAD2] [C]1 |2|1|1
$command_4$: No command to execute. Halt state.

*Step 7* (aligner):
45+67=112

### A.3.2 MULTIPLICATION

*Step 1* (aligner):
89*2=

*Step 2* (executor):
$state_0$: MUL, q0, [HEAD1]|9|8 [HEAD2]|2 [COUNT] [OUTPUT]
$command_0$: CMD [COUNT] 0, [OUTPUT] 0, q1

*Step 3-1, before call* (executor):
$state_1$: MUL, q1, [HEAD1]|9|8 [HEAD2]|2 [COUNT]|0 [OUTPUT]|0
$command_1$: CMD [CALL] LESS_THAN, q2
$callee\_state_0$: LESS_THAN, q0, [HEAD1] |0[HEAD2] |2 [OUTPUT]
$callee\_command_0$: CMD [HEAD1] RIGHT, [HEAD2] RIGHT, [OUTPUT] False, q1

*Step 3-1, after call* (executor):
$state_1$: MUL, q1, [HEAD1]|9|8 [HEAD2]|2 [COUNT]|0 [OUTPUT]|0
$command_1$: CMD [CALL] LESS_THAN, q2
$callee\_state_H$: LESS_THAN, qH, |0[HEAD1] |2[HEAD2] True
$callee\_command_H$: No command to execute. Halt state.

*Step 4-1, before call* (executor):
$state_2$: MUL, q2, [HEAD1]|9|8 [HEAD2]|2 [COUNT]|0 [OUTPUT]|0
$command_2$: CMD [CALL] ADD, q3
$callee\_state_0$: ADD, q0, [HEAD1] |9|8[HEAD2] |0 [C] [OUTPUT]
$callee\_command_0$: CMD: [C] 0, [HEAD1] RIGHT, [HEAD2] RIGHT, q1

*Step 4-1, after call* (executor):
$state_2$: MUL, q2, [HEAD1]|9|8 [HEAD2]|2 [COUNT]|0 [OUTPUT]|0
$command_2$: CMD [CALL] ADD, q3
$callee\_state_H$: ADD, qH, |9|8[HEAD1] |0[HEAD2] [C]0 |9|8
$callee\_command_H$: No command to execute. Halt state.

*Step 5-1, before call* (executor):
$state_3$: MUL, q3, [HEAD1]|9|8 [HEAD2]|2 [COUNT]|0 [OUTPUT]|9|8
$command_3$: CMD [CALL] ADD, q1
$callee\_state_0$: ADD, q0, [HEAD1] |0[HEAD2] |1 [C] [OUTPUT]
$callee\_command_0$: CMD: [C] 0, [HEAD1] RIGHT, [HEAD2] RIGHT, q1

*Step 5-1, after call* (executor):
$state_3$: MUL, q3, [HEAD1]|9|8 [HEAD2]|2 [COUNT]|0 [OUTPUT]|9|8
$command_3$: CMD [CALL] ADD, q1
$callee\_state_H$: ADD, qH, |0[HEAD1] |1[HEAD2] [C]0 |1

$callee\_command_H$: No command to execute. Halt state.

*Step 6-1, before call* (executor):
$state_4$: MUL, q1, [HEAD1]|9|8 [HEAD2]|2 [COUNT]|1 [OUTPUT]|9|8
$command_4$: CMD [CALL] LESS_THAN, q2
$callee\_state_0$: LESS_THAN, q0, [HEAD1] |1[HEAD2] |2 [OUTPUT]
$callee\_command_0$: CMD [HEAD1] RIGHT, [HEAD2] RIGHT, [OUTPUT] False, q1

*Step 6-2, after call* (executor):
$state_4$: MUL, q1, [HEAD1]|9|8 [HEAD2]|2 [COUNT]|1 [OUTPUT]|9|8
$command_4$: CMD [CALL] LESS_THAN, q2
$callee\_state_H$: LESS_THAN, qH, |1[HEAD1] |2[HEAD2] True
$callee\_command_H$: No command to execute. Halt state.

*Step 7-1, before call* (executor):
$state_5$: MUL, q2, [HEAD1]|9|8 [HEAD2]|2 [COUNT]|1 [OUTPUT]|9|8
$command_5$: CMD [CALL] ADD, q3
$callee\_state_0$: ADD, q0, [HEAD1] |9|8[HEAD2] |9|8 [C] [OUTPUT]
$callee\_command_0$: CMD: [C] 0, [HEAD1] RIGHT, [HEAD2] RIGHT, q1

*Step 7-2, after call* (executor):
$state_5$: MUL, q2, [HEAD1]|9|8 [HEAD2]|2 [COUNT]|1 [OUTPUT]|9|8
$command_5$: CMD [CALL] ADD, q3
$callee\_state_H$: ADD, qH, |9|8[HEAD1] |9|8[HEAD2] [C]1 |8|7|1
$callee\_command_H$: No command to execute. Halt state.

*Step 8-1, before call* (executor):
$state_6$: MUL, q3, [HEAD1]|9|8 [HEAD2]|2 [COUNT]|1 [OUTPUT]|8|7|1
$command_6$: CMD [CALL] ADD, q1
$callee\_state_0$: ADD, q0, [HEAD1] |1[HEAD2] |1 [C] [OUTPUT]
$callee\_command_0$: CMD: [C] 0, [HEAD1] RIGHT, [HEAD2] RIGHT, q1

*Step 8-2, after call* (executor):
$state_6$: MUL, q3, [HEAD1]|9|8 [HEAD2]|2 [COUNT]|1 [OUTPUT]|8|7|1
$command_6$: CMD [CALL] ADD, q1
$callee\_state_H$: ADD, qH, |1[HEAD1] |1[HEAD2] [C]0 |2
$callee\_command_H$: No command to execute. Halt state.

*Step 9-1, before call* (executor):
$state_7$: MUL, q1, [HEAD1]|9|8 [HEAD2]|2 [COUNT]|2 [OUTPUT]|8|7|1
$command_7$: CMD [CALL] LESS_THAN, q2
$callee\_state_0$: LESS_THAN, q0, [HEAD1] |2[HEAD2] |2 [OUTPUT]
$callee\_command_0$: CMD [HEAD1] RIGHT, [HEAD2] RIGHT, [OUTPUT] False, q1

*Step 9-2, after call* (executor):
$state_7$: MUL, q1, [HEAD1]|9|8 [HEAD2]|2 [COUNT]|2 [OUTPUT]|8|7|1
$command_7$: CMD [CALL] LESS_THAN, q2
$callee\_state_H$: LESS_THAN, qH, |2[HEAD1] |2[HEAD2] False
$callee\_command_H$: No command to execute. Halt state.

*Step 10* (executor):
$state_8$: MUL, qH, [HEAD1]|9|8 [HEAD2]|2 [COUNT]|2 |8|7|1
$command_8$: No command to execute. Halt state.

*Step 11* (aligner):
89*2=178

## A.4 IMPLEMENTATION OF SUBTRACTION OPERATOR

We implement subtraction in the CAEF framework by drawing inspiration from how subtraction is handled in CPUs. For subtraction in the form $a - b = c$, the process can be broken down into four steps:

1. Compute Reflection$(a, b)$: Generate a number $a_9$, where all digits are 9 and it is the same length as $a$. Perform a reflection operation, which is essentially subtraction, between $a_9$ and $b$. Since all digits of $a_9$ are 9, no borrowing occurs during this subtraction. Let the result of this step be $p$.

2. Compute $a + p$, and let the result be $q$.

3. Compute $q + 1$, and let the result be $r$.

4. Compute Left_mask$(r)$: Remove the leading 1 from the most significant digit of $r$. After this step, the final result, $c$, is obtained.

For example, in the case of $4531 - 1504 = 3027$, the process is as follows:

---

*Step 1* (Reflection):
Reflection$(4531, 1504) = 9999 - 1504 = 8495$

*Step 2* (Addition):
$4531 + 8495 = 13026$

*Step 3* (Addition):
$13026 + 1 = 13027$

*Step 4* (Left_mask):
Left_mask$(13027) = 3027$

---

In CAEF, steps 2 and 3 can be handled using the addition executor, which has already learned the logic for addition, while the auxiliary operators needed for steps 1 and 4 are relatively simple to implement. The subtraction executor composer only needs to learn how to sequentially invoke these basic executors to perform subtraction.

## A.5 PROMPTS USED IN BASELINE

Prompt used for LLaMA 3.1-8B pretrained model fine-tuned with LoRA:

---

*For addition, subtraction, multiplication, division:*
Please calculate the expression.
The expression is: {expr}.
The final answer should be presented in integer form!
Your output should be an integer.
The answer is: {response}

*For greater_than, less_than, equal:*
Please judge the expression is true or false.
The expression is: {expr}.
The final answer should be True or False!
Your output should be a word.
The answer is: {response}

---

Prompt used for LLaMA 3.1-8B-Instruct model:

> *For addition, subtraction, multiplication, division:*
> Please calculate the expression. The expression is: {expr}.
> The final answer should be presented in integer form.
> In your output, the final answer should be on its own line at the end, starting with 'Answer: '.
>
> *For greater_than, less_than, equal:*
> Please judge the expression is true or false. The expression is {expr}.
> The final answer that you give should be true or false.

Prompt used for GPT-4o:

> *For addition, subtraction, multiplication, division:*
> Please calculate the expression. The expression is: {expr}.
> The final answer should be presented in integer form.
> In your output, the final answer should be on its own line at the end, starting with 'Answer: '.
>
> *For greater_than, less_than, equal:*
> Please judge the expression is true or false. The expression is {expr}.
> The final answer that you give should be true or false.

### A.6 FURTHER EXPERIMENT RESULTS ANALYSIS

Using addition in the form of $a + b = c$ as an example, we generate the executor's training dataset by dividing the expressions into equivalence classes based on the pair $(\text{len}(a), \text{len}(b))$, where 20 random arithmetic expressions are generated for each equivalence class. When the operand lengths are sufficiently large, 20 samples are sparse across the entire equivalence class space. However, the model still achieves high accuracy in tasks such as 100-digit addition, indicating that the LLM effectively learns the logic of the Turing machine's transition function during training, thereby indirectly grasping the underlying logic of arithmetic computation.

However, this sampling strategy alone can lead to poorer performance when operand lengths are relatively short, typically less than 10 digits, compared to longer operands. We believe this issue arises because the longest samples in the training set generally exceed 100 digits, and from the perspective of equivalence classes, the dataset becomes dominated by samples with operands of several dozen digits. Intuitively, although both cases involve a difference of 10 digits, the difference between 5 and 15 digits has a much larger impact than the difference between 90 and 100 digits, especially in the way the LLM perceives these distinctions. Therefore, in practice, we slightly increase the number of samples from equivalence classes with shorter operands. Additionally, for operators such as ==, purely random sampling makes it difficult to obtain samples where the result is True, so some additional intervention is necessary.

### A.7 COMPUTATIONAL COMPLEXITY ANALYSIS

For the seven operators implemented using the CAEF framework, we assume the longer operand has a length of $d$. Based on the computation mechanism of self-attention, the computational complexity of a single model inference is $O(d^2)$.

For the *addition*, *greater_than*, *less_than*, and *equal* operators, the computation is performed digit by digit, requiring at most $d$ model queries for a complete computation. Additionally, the aligner performs two representation conversions, resulting in a total of $d + 2$ model queries. Therefore, the overall computational complexity is $O(d^3)$.

For *subtraction*, the computational complexity of the auxiliary operators *Reflection* and *Left_mask* is also $O(d^3)$. Since subtraction involves one call each to *Reflection* and *Left_mask*, along with two calls to *addition*, the overall complexity remains $O(d^3)$.

For *multiplication*, the situation is slightly more complex. For a calculation of the form $a \times b$, we assume $\text{len}(a) = d_1$, $\text{len}(b) = d_2$, and $\text{len}(a \times b) = d_3$. The number of iterations in the loop is $b+1$. During each iteration, *less_than* and two *addition* operations are performed, with the complexities as follows:

- For *less_than*, the longer operand has a length of $d_2$, resulting in a total complexity of $O((b+1)d_2^2)$ across all iterations.
- For the first *addition*, the longer operand has a length of $d_3$, giving a total complexity of $O((b+1)d_3^2)$ across all iterations.
- For the second *addition*, the longer operand again has a length of $d_2$, resulting in a total complexity of $O((b+1)d_2^2)$ across all iterations.

Thus, the overall complexity is the sum of these three parts, plus the two aligner conversions. The final computational complexity for multiplication is $O(bd_2^2 + bd_3^2)$.

For *division*, the situation is similar to multiplication. Assuming $\text{len}(a) = d_1$, $\text{len}(b) = d_2$, $a \div b = c$, and $\text{len}(c) = d_3$, the number of iterations in the loop is $c+1$. The final computational complexity for division is $O(cd_1^2 + cd_3^2)$.

## A.8 ATTEMPTS TO MERGE ALIGNER AND EXECUTOR

We attempt to combine the functionalities of the aligner and executor into one LoRA adapter. Table 4 shows the experimental results we obtained in the addition operator:

Table 4: Comparison of the results for merging the aligner and executor on the addition operator with the original CAEF method. The left side of the slash shows the results after merging the executor and aligner, while the right side presents the original results of CAEF.

| Setting | 5-digits | 10-digits | 50-digits | 100-digits |
|---|---|---|---|---|
| executor & aligner | 90.3/100.0 | 97.3/99.6 | 94.9/99.9 | 90.0/98.6 |
| executor | 100.0/100.0 | 99.9/100.0 | 99.6/99.9 | 97.4/99.6 |
| aligner (I) | 90.3/100.0 | 97.5/99.7 | 95.7/100.0 | 94.0/99.6 |
| aligner (O) | 100.0/100.0 | 99.9/99.9 | 99.5/100.0 | 98.0/99.4 |

From the results, we observe the following:

- The performance of the executor and aligner (O) shows a slight degradation compared to the original modular approach in most of the experimental settings.
- The aligner (I), however, experiences a significant performance drop. As a result, merging the aligner and executor leads to a substantial decline in the overall accuracy of addition.

Additionally, merging these two components introduces the challenge of determining the appropriate ratio for training samples from both parts. Therefore, based on the experimental results, we believe separating the executor and aligner remains the preferable approach.

