# OpenReview forum: "Executing Arithmetic: Fine-Tuning Large Language Models as Turing Machines"
_ICLR.cc/2025/Conference — Submitted to ICLR 2025_

### Official Review · Reviewer_F4pL · 2024-10-29

**Soundness:** 1
**Presentation:** 2
**Contribution:** 1
**Rating:** 5
**Confidence:** 3

**Summary:**

The paper proposes a new framework, called CAEF (Composable Arithmetic Execution Framework), for enabling large language models (LLMs) to perform arithmetic computations. The framework uses the model to emulate a Turing machine, where an input problem (e.g. 89+2) is solved through a series of <state, command> pairs. States and commands represented in a string format are fed to the model, which performs the computation (transitions) with trained modules for input/output alignment and operation execution. These modules are implemented as LoRA adapters, which are trained separately for different operations, (e.g., there are different adapters for addition and multiplication). Experiments are conducted on LLaMA 3.1, where the framework performance is evaluated per operation in comparison to (a) LLaMA 3.1 fine-tuned with LoRA adapters in an end-to-end fashion, (b) the instruction-tuned version of LLaMA 3.1, and (c) GPT-4o. CAEF often shows substantial gains in performance compared to the baselines, mostly in problems involving numbers with over 10 digits.

**Strengths:**

* The proposed framework is interesting and creative.

* The results show substantial improvements for numbers with >10 digits.

**Weaknesses:**

* While the paper is very interesting, I found it hard to digest as I was struggling to understand the proposed framework and exactly how it is being implemented. Specifically, while I was reading I felt like everything is described on an abstract level, without any grounding in concrete examples or settings. Only at the end of the implementation section (3.3) and after looking at the examples in the appendix, I saw some concrete examples of how the framework is executed. It will be helpful to be more explicit and clear about what a state is, what a command is, what the input and output from the models are exactly, whether there is a single execution of the model or multiple inference passes, etc. There are descriptions of what a state should include and in sections 3.2 and 3.3 there are illustrations for specific examples, but the illustrations are abstract and the explanation is vague.

* If I understand correctly, given a problem (e.g. 89 times 2), the model is executed multiple times, where every transition corresponds to a single inference pass. If this is correct, then I am not sure if the experiments make a fair comparison with the baselines. CAEF executes the model many times using a fine-grained step-by-step solution, while the other baselines solve the problem in an end-to-end manner. Looking at the prompts in the appendix, it seems that the instruct models (e.g. GPT) were not provided with instructions to perform chain-of-thought reasoning or anything that would encourage them to solve the problem step by step.

* There is something problematic about the premise of the paper and specifically the focus on exact arithmetic computations of large numbers (with >10 digits). Why do we need LLMs to perform this kind of computation? Wouldn’t an LLM with an external tool be a better solution in this case? Also related to this point is that one potential argument for an LLM-based solution would be that the LLM could infer the type of problem described and the steps for solving it. However, if I understood correctly, with the proposed framework the model does not make this kind of inference. Namely, the adapters to be used are selected in advance. I may have missed this point, tried to look in the paper but couldn’t find it indicated explicitly anywhere. I assume that the adapters are pre-selected (not by the model) because the results are presented per operator and there are no reported results for the performance on inferring the right adapters for the problem.

**Questions:**

* Missing reference:
Giving BERT a Calculator: Finding Operations and Arguments with Reading Comprehension, EMNLP 2019.

* Figures are very small and hard to read. Consider showing explicitly what the input and output from the model are.

* Line106: incorrect citation format.

---

> ### Author Response · Authors · 2024-11-22
> **Response to Reviewer F4pL**
>
> ## W1: Clarity of Framework and Implementation
>
> Thank you for your helpful suggestions regarding the presentation of the paper. In response to your comment, we will revise the text in the Introduction and explicitly emphasize in Figure 1 that "executing computations in the CAEF framework requires multiple LLM queries." Additionally, we will restructure the paper to include concrete examples earlier on, which should make our approach easier to understand. We will also revise Figure 3 in Section 3 to clearly illustrate the inputs and outputs.
>
> ## W2: Comparison with Baselines
>
> Thank you for your thoughtful question; your understanding is correct and valid. We have conducted extended experiments to evaluate the baseline model’s performance under a CoT setup, as well as in a Turing machine-style computation similar to the CAEF framework. Details and results can be found in the global rebuttal section. As we can see, the results indicate that employing the single-query approach leads to a significant decline in model performance as the operand length increases. Additionally, due to the limitation of the context window, this method is incapable of handling scenarios involving operands with more than 50 digits. In contrast, adopting the multi-query approach without fine-tuning the model essentially fails to function effectively.
>
> ## W3.1: Need for LLMs in Exact Arithmetic Computations
>
> Thank you for your question. At this stage, we view our work as a methodological exploration rather than an application to specific real-world scenarios. A more detailed discussion of this issue is provided in the global rebuttal section.
>
> ## W3.2: Pre-selection of Adapters
>
> Thank you for your question. For a given operator, we are currently using a pre-selection method. However, there are existing methods that combine MoE architecture with LoRA, allowing automatically switching LoRA adapters. These methods could potentially address this issue. A more detailed discussion of this approach can be found in the global rebuttal section.
>
> ## Questions
>
> Thank you for pointing out the writing issues in the paper. We will correct these in the revised version.

---

> > ### Comment · Reviewer_F4pL · 2024-11-25
> >
> > Thanks, authors, for responding to my concerns and questions.
> >
> > Regarding W1 (clarity): Thanks for considering my suggestions, I think making these revisions and adding the clarifications will make the paper easier to understand.
> >
> > Regarding W2 and the new results: Which operands were considered in these experiments? It is hard to understand how these new results compare to the numbers reported in the paper. Specifically, the performance for LLaMA 3.1(I) seems to be very low, much lower compared to its performance without CoT (across all operands) which is a bit surprising.

---

> > > ### Author Response · Authors · 2024-11-25
> > > **Response to Reviewer F4pL**
> > >
> > > Thank you very much for your new questions. We are currently revising the paper and will submit the updated version as soon as possible. We expect to complete the revisions by tomorrow.
> > >
> > > Regarding the new questions:
> > >
> > > 1. Operands are generated through uniform sampling. For example, in the case of 5-digit addition, we sample two numbers independently from the range \\([10,000, 99,999]\\) to obtain the operands.
> > >
> > > 2. Under the "detailed scratchpad" CoT prompt, the model needs to decompose the operands during the calculation process. Based on our observations, the LLaMA 3.1-8B Instruct model often makes mistakes in selecting the corresponding digits during step-by-step computation. For instance, when calculating \\(12345 + 67890\\), the model frequently errs in choosing the digits \\(3\\) and \\(8\\) at the third position.
> > >
> > >    For the LLaMA 3.1 series models, all combinations of 1-, 2-, and 3-digit numbers are tokenized. Taking \\(12345\\) as an example, it is tokenized into two tokens: \\(123\\) and \\(45\\). Consequently, splitting these tokens into \\(1\\), \\(2\\), \\(3\\), \\(4\\), and \\(5\\) requires additional effort, making the decomposition process more challenging than the computation itself. We believe that this difficulty leads to significantly lower accuracy.
> > >
> > >    This observation aligns with the conclusion in our paper that "in CAEF, the bottleneck in computation accuracy lies in the reversal of operands rather than the computation itself." Enhancing the model's ability to decompose digits requires extra fine-tuning, as models struggle to perform such tasks effectively out-of-the-box.
> > >
> > > We hope this addresses your concerns and provides clarity. Please let us know if there are further questions or points you would like us to elaborate on.

---

> > > > ### Comment · Reviewer_F4pL · 2024-11-28
> > > >
> > > > Thanks for the clarification. Could you please also clarify which *operator(s)* were considered in the new experiments?

---

> > > > > ### Author Response · Authors · 2024-11-28
> > > > >
> > > > > Thank you very much for your response. The new experiments we conducted are on the addition operator. We apologize for not making this clear in the global rebuttal earlier, and we have now revised the global rebuttal to clarify this point.
> > > > >
> > > > > Due to time constraints, we have only completed a full set of experiments on the addition operator so far. However, based on our current observations, the results for the other operators appear to be similar. In the next revision, we plan to complete experiments for all seven operators and include the results.

---

> > > > > > ### Comment · Reviewer_F4pL · 2024-11-28
> > > > > >
> > > > > > Thanks for the response. Given the authors' clarifications and additional results, I have increased my score from 3 to 5. While I am still concerned about the motivation for this kind of work in the era of tool usage in LLMs, I understand the authors' point of view and appreciate the exploration of diverse research directions. That said, I believe that improving the clarity of the paper (as the authors started doing) and extending and adding the CoT experiments are necessary for meeting the ICLR acceptance bar.

---

> ### Comment · Area_Chair_NbbJ · 2024-11-25
> **[Reminder] Response to Authors**
>
> Dear Reviewer,
>
> As the rebuttal period is drawing to a close, I would appreciate your response to the authors' rebuttal at your earliest convenience.
>
> Best Regards,
>
> Area Chair

---

### Official Review · Reviewer_WNj4 · 2024-10-31

**Soundness:** 3
**Presentation:** 3
**Contribution:** 2
**Rating:** 6
**Confidence:** 4

**Summary:**

This study proposes to decompose basic arithmetic operations into smaller steps that can be executed by LLMs simulating Turing machines. Specifically, by adapting LLMs into separate aligner and executor modules (both via low-rank adapters), one can map an arithmetic operation into a series of smaller atomic steps that solve the task. At each timestep, the aligner translates the input into a representation that the executor can process. The training data for the aligner and executor are obtained by simulating Turing machine prototypes.

The proposed approach achieves significantly higher accuracies than base LLMs, and generalizes better to inputs with many digits.

**Strengths:**

* The proposed approach performs significantly better than base LLMs on basic arithmetic operations.
* At a high level, I admire the ambition of implementing computational-theoretic data structures using LLMs. If this works well in a wider variety of settings, it could be applied as a natural-language interface to theoretical computational structures that do not require significant human effort to construct. This could have benefits in both educational and software engineering settings, especially if the LLM is capable of explaining each step of its processing.
* The modularity of the aligner and executor setup may make it easier to debug specific components in cases where the approach does not perform well.
* The idea is mostly clearly presented.

**Weaknesses:**

* To generate training data, one needs to create a Turing machine prototype. But once one has this machine, why not just use that to solve the task instead of having an LLM do it? It seems like one is better off having the LLM call the machine as a tool, rather than adapting the LLM to execute this procedure. The one way I could see this approach being beneficial over just using the machine is if the LLM can generalize to harder instances than the original machine was ever designed to handle, or to harder instances than were in the generated training data.
* Basic arithmetic operations represent a relatively easy application. I don’t think this is a significant weakness, since this is otherwise a pretty convincing proof-of-concept, but it may limit the impact of this otherwise neat idea.
* The modularity of the aligner and executor setup means that one must maintain high performance for multiple components over many timesteps. This means that a single point of failure could lead to cascading errors on harder tasks.
* Terminology like “understanding” is abused throughout the paper. In particular, the claim of “general understanding” in the abstract is a significant overclaim, and uses of “understanding” afterwards (especially at L189) are unnecessary and anthropomorphize LLMs. The paper would feel much more objective without this language.
* More of a suggestion than a weakness: the error analysis alluded to in the Limitations section sounds interesting. This would be great as its own Error Analysis section.

**Questions:**

Questions
===
1. What is the advantage of using LLMs to simulate the required Turing machine rather than simply using the Turing machine?
2. Rather than using separate aligner and executor modules, is it possible to perform each step in an end-to-end manner (i.e., using just one adapted LLM to do an alignment and execution step)? This could cut the number of possible points of failure in half.

Typos/Suggestions
===
* L50-51: citation or forward reference needed
* L106: \citet -> \citep
* L110-111: run-on
* L133: some examples/citations would be appreciated
* L183: nitpick: The representation isn’t the space in which the computation takes place. The representation is merely a mathematical abstraction that makes it easier for humans to understand.
* The “step” variable is sometimes presented in text mode, and sometimes in math mode. It would be nice to standardize notation.
* L358: small suggestion: “original” -> “unmodified”. “Original” is polysemous, and the sense of “new” interfered with the intended sense of “unmodified” on my first pass

---

> ### Author Response · Authors · 2024-11-22
> **Response to Reviewer WNj4**
>
> ## W1: Use of Turing Machine
>
> Thank you for your question. One of the key assumptions of our work is that **the model does the reasoning by itself, without using any external tools**. At this stage, we view our work as a methodological exploration rather than an application to specific real-world scenarios. A more detailed discussion of this point can be found in the global rebuttal section.
>
> ## W2: Basic Arithmetic Operations as Application
>
> Thank you for your valuable suggestion. We share a similar perspective on this matter. While arithmetic tasks may appear straightforward, they are highly suitable for demonstrating the core principles of our approach. Our experimental results highlight that current LLMs struggle to achieve high accuracy in tasks involving large numbers. The proposed CAEF framework represents a foundational step towards "System 2" reasoning (as discussed in the global rebuttal section), specifically focusing on training LLMs to perform executing tasks. Moving forward, one of our ongoing work is to extend CAEF by incorporating search capabilities into LLMs to tackle more complex problems, particularly in the domain of automated theorem proving.
>
> In the domain of automated theorem proving, existing methods, which rely on proof-step generation (each proof step is generated sequentially), often depend on external tools (e.g., Lean 4) to validate the correctness of individual steps and to estimate the optimal next step using search algorithms. We propose extending the CAEF approach to this domain, where models generating individual proof steps serve as the basic executors, and models responsible for devising search strategies act as the executor composers. By refining proof steps and providing richer contextual information throughout the proof process, we aim to enhance both proof generation quality and search efficiency. This approach aligns closely with the "System 2" reasoning paradigm outlined in the global rebuttal section.
>
> ## W3: Risk of Cascading Errors
>
> Thank you for your question. As shown by the experimental results in the paper and the extended results provided in the global rebuttal section, our approach does introduce cascading errors due to inaccuracies in individual steps. However, we mitigate this by achieving sufficiently high accuracy at each step through task simplification. For example, in the case of 100-digit addition, the executor's step accuracy needs to exceed 0.9999 to ensure an overall accuracy of 0.99. Despite the cascading errors, our results consistently outperform end-to-end generation methods.
>
> ## W4: Misuse of Terminology: "Understanding"
>
> Thank you for pointing out the inappropriate use of the term “understanding.” We acknowledge this issue and will revise the expression to a more suitable one in the revised version of the paper.
>
> ## W5: Suggestion for Error Analysis Section
>
> Thank you for your suggestion. We have added an analysis of this issue in the global rebuttal section. Please refer to it for further details. The analysis will also be updated in the revised version of the paper
>
> ## Q1
>
> Thank you. This issue is the same as the one raised in W1. Please refer to our response to W1 for further clarification.
>
> ## Q2
>
> Thank you. We conducted an additional experiment to investigate merging the aligner and executor into a single module using one LoRA adapter for the addition operator. The evaluation was performed on the same test set as in the paper, and the results are as follows:
>
> |                    |  5-digits   | 10-digits  | 50-digits  | 100-digits |
> | ------------------ | :--------:  | :--------: | :--------: | :--------: |
> | executor + aligner | 90.3/100.0  | 97.3/99.6  | 94.9/99.9  | 90.0/98.6  |
> | executor           | 100.0/100.0 | 99.9/100.0 | 99.6/99.9  | 97.4/99.6  |
> | aligner(I)         | 90.3/100.0  | 97.5/99.7  | 95.7/100.0 | 94.0/99.6  |
> | aligner(O)         | 100.0/100.0 | 99.9/99.9  | 99.5/100.0 | 98.0/99.4  |
>
> The left side of slash shows the results after merging the executor and aligner, while the right side presents the original results of CAEF. From the results, we observe the following:
>
> * The performance of the executor and aligner (O) shows a slight degradation compared to the original modular approach.
>
> * The aligner (I), however, experiences a significant performance drop.
>
> Under this setup, the bottleneck of the computation remains the reversal of operands rather than the computation itself, which aligns with the conclusions presented in our paper. Based on these findings, we believe separating the executor and aligner remains the preferable approach.
>
> In the **"Application of CAEF to Math Word Problems"** section of the global rebuttal, we further discuss potential solutions for integrating these modular LoRA adapters. This could address the issues raised and provide a pathway to improve performance.
>
> ## Suggestions
>
> Thank you for pointing out the writing issues in the paper. We will correct these in the revised version.

---

> > ### Comment · Reviewer_WNj4 · 2024-11-24
> > **Response**
> >
> > Thank you for the detailed response! Weakness 3 has been addressed. It sounds like W1 and W2 will be left to future work. W4 has not yet been addressed in the PDF, but hopefully this can be fixed quickly. W5 has been discussed in the global rebuttal; it is unfortunate that no further trends were obvious, but I appreciate the authors' looking into it.
> >
> > Thanks also for the investigation of Q2! These results are interesting, and I think showing this in the paper (even just an appendix) will make it easier to justify using the two-module method.
> >
> > I have read the other reviewers' reviews, and agree that the proposed method is not particularly efficient nor practical compared to other tools that can perform arithmetic. However, I still think this is an interesting proof-of-concept with surprisingly good results, especially considering the potential for cascading errors. The experimentation also seems largely well done. Given this and the above feedback, I am choosing to keep my score at a borderline accept. I think this would be valuable to the community as an idea, but probably not particularly useful or widely impactful until some of these practical issues are either resolved, or some stronger theoretical argument for the value of this system is presented.

---

> > > ### Author Response · Authors · 2024-11-25
> > > **Response to Reviewer WNj4**
> > >
> > > Thank you very much for your response. We are currently revising the paper and will submit the updated version as soon as possible. We expect to complete the revisions by tomorrow.

---

### Official Review · Reviewer_CdN9 · 2024-11-04

**Soundness:** 2
**Presentation:** 3
**Contribution:** 1
**Rating:** 3
**Confidence:** 4

**Summary:**

The paper attempts to address the low performance of LLMs such as llama-3.1 in arithmetic tasks such as addition and multiplication. They have devised a text-based Turing machine for these tasks, and have fine-tuned an LLM to execute it.

**Strengths:**

1- Paper's approach in solving arithmetic tasks is grounded in our concrete knowledge of Turing machines, which is less ubiquitous in the literature of length generalization.

2- Paper is well-written and every part has come with specific goal. Sections 2 to 4 follow a natural recipe to explain the method, along with the results.

**Weaknesses:**

1- In the abstract, from line 16 to 18, authors claim that their method enables LLMs to execute arithmetic tasks in a more rigorous manner and achieve better generalization. My initial understanding of the passage was that the execution happens in a single query of the LLM, where the output includes the final answer. Whereas in Section 3.2, one realizes that a query to the LLM accounts to running a single step of their control flow.

2- That said, I'm not convinced the proposed approach can be the primary solution for arithmetic tasks. If I understand correctly, the number of queries increases proportionally to the length of the operands in the task, which is problematic since every query's needed number of auto-regression steps also grows linearly with the length of operands, leading to O(d^2) number of auto-regression steps compared to O(d) for a generating the answer with a decoder-only model (d is the length of operands). The computational analysis is missing in the paper.

3- In section 4, it's not clear whether the comparison with typical LLMs is done by executing the explained Turing machine with each of them and running the steps in different streams, or asking the answer in a single query to these models. I think a comparison with both methods is necessary when reporting.

4- To my understanding, the proposed method uses two modules for the implementation: "BASIC EXECUTORS" and "EXECUTOR COMPOSERS". And for working with each of the arithmetic tasks (+, −, ×, ÷, ==, >, and <) they utilize different low rank adapters for fine-tuning. This doesn't give a generic solution for solving arithmetics at test time, where based on the given sample a collection of modules must be placed after each other.

5- How does the method perform in the presence of text? I do not see any sections discussing this.

**Questions:**

Please address the concerns expressed above.

---

> ### Author Response · Authors · 2024-11-22
> **Response to Reviewer CdN9 W1 to W4**
>
> ## W1: Clarification of Execution Process
>
> Thank you for your comment regarding the expression in the paper. Your understanding is correct. We will revise the expression in the Introduction to clarify this point and explicitly highlight in Figure 1 that "executing computations in the CAEF framework requires multiple LLM queries."
>
> ## W2: Computational Complexity
>
> Thank you for your suggestion. For the seven operators implemented using the CAEF framework, we assume the longer operand has a length of \\(d\\). Based on the computation mechanism of self-attention, the computational complexity of a single model inference is \\(O(d^2)\\).
>
> For the *addition*, *greater_than*, *less_than*, and *equal* operators, the computation is performed digit by digit, requiring at most \\(d\\) model queries for a complete computation. Additionally, the aligner performs two representation conversions, resulting in a total of \\(d + 2\\) model queries. Therefore, the overall computational complexity is \\(O(d^3)\\).
>
> For *subtraction*, the computational complexity of the auxiliary operators *Reflection* and *Left_mask* is also \\(O(d^3)\\). Since subtraction involves one call each to *Reflection* and *Left_mask*, along with two calls to *addition*, the overall complexity remains \\(O(d^3)\\).
>
> For *multiplication*, the situation is slightly more complex. For a calculation of the form \\(a \times b\\), we assume \\(\text{len}(a) = d_1\\), \\(\text{len}(b) = d_2\\), and \\(\text{len}(a \times b) = d_3\\). The number of iterations in the loop is \\(b + 1\\). During each iteration, *less_than* and two *addition* operations are performed, with the complexities as follows:
>
> - For *less_than*, the longer operand has a length of \\(d_2\\), resulting in a total complexity of \\(O((b+1)d_2^2)\\) across all iterations.
> - For the first *addition*, the longer operand has a length of \\(d_3\\), giving a total complexity of \\(O((b+1)d_3^2)\\) across all iterations.
> - For the second *addition*, the longer operand again has a length of \\(d_2\\), resulting in a total complexity of \\(O((b+1)d_2^2)\\) across all iterations.
>
> Thus, the overall complexity is the sum of these three parts, plus the two aligner conversions. The final computational complexity for multiplication is \\(O(bd_2^2 + bd_3^2)\\).
>
> For *division*, the situation is similar to multiplication. Assuming \\(\text{len}(a) = d_1\\), \\(\text{len}(b) = d_2\\), \\(a \div b = c\\), and \\(\text{len}(c) = d_3\\), the number of iterations in the loop is \\(c + 1\\). The final computational complexity for division is \\(O(cd_1^2 + cd_3^2)\\).
>
> We will include this analysis in the revised version of the paper.
>
> ## W3: Comparison in Section 4
>
> Thank you for your question. The baseline setup involves the LLM generating results in a single-query mode. We completely understand your concern, and in response, we have conducted extended experiments comparing the model's performance under a CoT setup and a Turing machine-style computation, similar to the CAEF framework. Details and results can be found in the global rebuttal section. As we can see, the results indicate that employing the single-query approach leads to a significant decline in model performance as the operand length increases. Additionally, due to the limitation of the context window, this method is incapable of handling scenarios involving operands with more than 50 digits. In contrast, adopting the multi-query approach without fine-tuning the model essentially fails to function effectively.
>
> ## W4: Modular Approach and Generic Solution
>
> Thank you for your insightful suggestion. We completely agree with your concern. Initially, our idea was to design the LLM to imitate a universal Turing machine. However, we decided to pursue the current approach for two key reasons:
>
> 1. **Choice of LoRA for Fine-Tuning**: We selected LoRA as the fine-tuning method due to its lightweight nature. However, this approach has limited capacity for injecting knowledge into the model. Using a single LoRA adapter to encode all operators’ knowledge proved challenging. While full model fine-tuning might be an alternative, it comes with its own set of trade-offs.
>
> 2. **Why LoRA Over Full Model Fine-Tuning**: As noted in the paper, our goal is to enable the model to build upon previously learned operators while incorporating new computation logic. Full fine-tuning requires incremental training for new operators, which, if the logic is orthogonal to existing operators, could lead to catastrophic forgetting. Distributing knowledge across separate lightweight LoRA adapters provides a more modular and efficient approach.
>
> Furthermore, there are emerging methods that combine MoE (Mixture of Experts) architecture with LoRA, which may offer potential solutions to this issue. A more detailed discussion about this can be found in the global rebuttal section.

---

> ### Comment · Area_Chair_NbbJ · 2024-11-25
> **[Reminder] Response to Authors**
>
> Dear Reviewer,
>
> As the rebuttal period is drawing to a close, I would appreciate your response to the authors' rebuttal at your earliest convenience.
>
> Best Regards,
>
> Area Chair

---

> ### Author Response · Authors · 2024-11-25
> **Response to Reviewer CdN9 W5**
>
> ## W5: Performance in the Presence of Text
>
> Thank you for your question. At present, our method does not directly address text-based tasks such as math word problems. However, in our vision, with the CAEF plugin, an LLM could seamlessly switch to CAEF for handling arithmetic calculations while generating reasoning steps for such problems. A more detailed discussion on this can be found in the global rebuttal section. This approach aligns with the response provided to W4.

---

### Official Review · Reviewer_sUkz · 2024-11-05

**Soundness:** 2
**Presentation:** 3
**Contribution:** 2
**Rating:** 5
**Confidence:** 4

**Summary:**

The paper addresses the limitations of Large Language Models (LLMs) in performing arithmetic tasks, where they often memorize examples rather than grasping the underlying computational logic. To overcome this, the authors propose the Composable Arithmetic Execution Framework (CAEF), which enables LLMs to perform step-by-step computations by simulating Turing Machines. This approach helps the models develop a true understanding of computational logic. Additionally, CAEF's scalability allows it to combine learned operators to ease the learning process for complex operations. The framework demonstrated nearly 100% accuracy across seven common mathematical operations using the LLaMA 3.1-8B model, including computations involving operands with up to 100 digits. This performance surpasses that of GPT-4o in certain aspects.

**Strengths:**

1. The paper is clearly presented and supplemented with ample illustrative figures, allowing readers to easily grasp its content.
For instance, Figure 1 shows the process of CAFE executing multiplication, Figure 2 outlines the CAFE framework, and Figures 3 and 4 detail the execution processes of addition and multiplication, respectively.

2. This paper addresses a critic issue in real-world applications of LLMs. The arithmetic performance of existing LLMs, whether open-source or proprietary, is often unreliable, making it difficult to trust their output.

3. The idea presented in the paper is refreshing, as converting seven common arithmetic operations into text language interpretable by LLMs is no easy feat. This is achieved using lora-adapters to build executors and aligners, relying on two specific prompts: state and command.

4. The main table presents great performance results for the seven types of arithmetic operations. The accuracy exceeds 99% for both 50-digit and 100-digit tasks, demonstrating practical applicability.

**Weaknesses:**

1. It is unfortunate that, as mentioned in Section 6 on limitations, the **efficiency issues**—stemming from *the repeated calls to model.generate()* and the *storage overhead of intermediate prompts and adapter representations*—make this approach challenging to deploy in practical applications.

2. When faced with repeating digit patterns such as "999..." or "456456...", the CAFE's executor and aligner fail to function properly.
I disagree with the authors' claim that this is an inherent limitation of LLMs. I believe it is more likely due to **the absence of such patterns in the fine-tuning data for the executor and aligner**.
This raises concerns about the robustness of the experimental results and *whether CAFE can consistently handle or fail in other unique digit patterns*.

**Questions:**

Please check Weakness.

---

> ### Author Response · Authors · 2024-11-22
> **Response to reviewer sUkz**
>
> ## W1: Efficiency issue
>
> Thank you for your thoughtful question. Your concerns are completely valid. At this stage, we consider our work to be a methodological exploration rather than a direct application to specific real-world scenarios. Addressing the efficiency issue is one of our ongoing efforts. For a more detailed discussion on this topic, please refer to the global rebuttal section.
>
> ## W2: CAEF may fail in some unique digit patterns
>
> Thank you for your insightful comment. We fully agree with your concern, as it aligned with our initial hypothesis when we first encountered this issue. However, after conducting a more detailed investigation, we arrived at the conclusion presented in the paper. For a more comprehensive discussion, please refer to the global rebuttal section.

---

> ### Comment · Reviewer_sUkz · 2024-12-01
> **Thanks for your rebuttal**
>
> While I appreciate your approach to conducting cutting-edge research, the issues present in the current proposal (also noted by other reviewers) cannot be overlooked.
>
> Specifically, there are concerns about how to handle queries containing text, the cost associated with making multiple queries to the LLM, and whether modifying the network architecture might impact the original foundational capabilities of the LLM.
>
> Therefore, my concerns about this paper remain unresolved, and I have decided to lower my score from 6 to 5.

---

### Author Response · Authors · 2024-11-22
**Global rebuttal section -- part 1**

We would like to thank all the reviewers for their their thoughtful and constructive comments and suggestions.

## Methodology for CAEF

First, let us explain the methodology behind our paper. In our vision, the ability to execute computational logic is a fundamental skill for LLMs. To illustrate this concept, we draw inspiration from Daniel Kahneman’s book *Thinking, Fast and Slow* (2011), which introduces two modes of thought: "System 1" and "System 2." "System 1" is fast, instinctive, and automatic, while "System 2" is slower, more deliberate, and logical. For example, solving a simple problem like \\( 2 + 2 \\) typically relies on "System 1" processing, whereas tackling a complex computation like \\( 329,475,349,23 + 238,963,435,739 \\) (similar to the evaluation tasks in our study) requires "System 2" reasoning.

Currently, most LLMs predominantly operate in a manner analogous to "System 1," generating responses based on learned patterns and intuitive fluency. In contrast, our proposed framework, CAEF, introduces a methodology for enabling "System 2" generation in LLMs. This allows models to handle complex tasks that require executing intricate logical rules or searching for optimal solutions (an aspect of our ongoing work), thereby significantly expanding their problem-solving capabilities.

## Clarification Regarding the Use of External Tools and Efficiency Issues

Our approach serves as a methodological exploration aimed at investigating the upper limits of a model's capabilities in arithmetic tasks without relying on external tools. At this stage, our primary focus is on enabling the model to imitate the operations of a Turing machine to achieve high computational accuracy, rather than on applying the method directly to specific downstream applications. However, we acknowledge that the need for multiple calls to `model.generate()` for each computation introduces significant efficiency challenges. Relying on external tools remains a more practical solution for real-world applications at this time.

Alternatively, from another perspective, achieving high accuracy in arithmetic tasks with current generative LLMs, without leveraging external tools, inherently incurs additional time overhead. A comparable case is OpenAI o1 LLM, which, while seemingly producing results in a single inference for users, likely employs internal methods such as MCTS-style search. This process, though transparent to the user, involves multiple steps and incurs substantial time costs.

To address these challenges, we are exploring potential solutions:

1. **Batch Output Generation in a Single Decoding Step**: One potential solution involves modifying the model's architecture to generate a batch of outputs in a single decoding step. In our case, the differences between consecutive computational states \\(s\_i\\) and \\(s\_{i+1}\\) are typically minor, often involving only token reordering. We believe this adjustment could significantly improve efficiency in our specific scenario.

2. **LLM-Friendly Random Access Memory (RAM) Model**: Another interesting research direction involves designing a memory model for LLMs. Currently, LLMs operate like CPUs, executing commands by reading and writing variables encoded as input/output strings. If an LLM-friendly RAM model were developed, the LLM could directly read/write from memory and execute operations only on relevant variables or digits. This would drastically reduce computational costs and improve efficiency.

We believe these directions are promising for addressing the efficiency limitations of our current approach and may also open up broader possibilities for enhancing LLM capabilities in computational tasks.

---

> ### Author Response · Authors · 2024-11-22
> **Global rebuttal section --- part 2**
>
> ## Application of CAEF to Math Word Problems
>
> First, we would like to clarify that, at this time, our method requires the manual selection of the active LoRA adapter, rather than enabling the model to autonomously select it. This limitation currently prevents the direct application of our method to solving math word problems.
>
> However, our approach can be viewed as a plug-in component. There are several studies (refer to \[1]) that focus on integrating Mixture of Experts (MoE) and LoRA techniques to enable automatic selection and switching of active LoRA adapters based on the input. These studies are orthogonal to our method, and we believe that combining these techniques with our approach would allow it to be effectively applied to math word problems. For instance, using the CAEF plug-in, an LLM could switch to CAEF for handling arithmetic calculations as part of the reasoning steps in solving a math problem.
>
> ## Discussion on Model Errors and Error Patterns
>
> We would like to provide further clarification regarding the “Prone to errors with repeated digit patterns” part. Our sampling methods for both the training and testing sets are consistent, ensuring that similar patterns are present in the training set. We have checked and verified this. Beyond this specific pattern, we have not identified any significant commonalities in the remaining error-inducing samples.
>
> We attribute this issue to the internal mechanisms of generative models. A common failure pattern during inference involves the model repeatedly generating the same segment of text. This behavior suggests that transformer-based models may have difficulty tracking how much of a sequence has already been repeated, leading to the continuation of the same string. Consequently, we hypothesize that inputs with repeated patterns exacerbate the model's fragility, increasing the likelihood of inconsistencies between the input and the generated outputs.
>
> ## **Extended Evaluation**
>
> To address the issue of fairness in comparing CAEF with the baselines, we conducted two additional experiments on the addition operator:
>
> 1. **CoT Group**: We used the "detailed scratchpad" CoT prompt in \[2] to evaluate three models: a LoRA-fine-tuned LLaMA 3.1 base model, LLaMA 3.1 Instruct model, and GPT-4o (consistent with the baseline in the paper). For this setup, the calculation results were obtained using a single-query approach.
>
> 2. **CAEF-style Group**: In this experiment, we instructed the LLaMA 3.1 Instruct model and GPT-4o to follow a CAEF-style multi-query approach. A 4-shot prompt was used to provide detailed Turing machine-style descriptions and examples.
>
> The evaluations were conducted on the same test set as in the paper, and the results are summarized below:
>
> 1. **CoT Group Results**:
>
> |              | 5-digits | 10-digits | 50-digits | 100-digits |
> | ------------ | :------: | :-------: | :-------: | :--------: |
> | LLaMA 3.1(L) | 100.0    | 63.5      | -         | -          |
> | LLaMA 3.1(I) | 23.3     | 1.5       | -         | -          |
> | GPT-4o       | 98.5     | 87.0      | -         | -          |
> | CAEF         | 100.0    | 99.6      | 99.9      | 98.6       |
>
> As the results shown in the Table, it is evident that the single-query approach, even when enhanced with a detailed scratchpad prompt containing explicit computational reasoning, struggles to handle scenarios with operands of 10 or more digits effectively. Additionally, because the detailed scratchpad method requires embedding all computational steps into a single query, context lengths can exceed 10k tokens for operands of 50 digits. This not only makes the method infeasible in large-number scenarios but also challenges the model’s ability to process such long contexts, even when supported by larger context windows.
>
> 2. **CAEF-style Group Results**:
>
> In the CAEF-style group, neither the LLaMA 3.1 Instruct model nor GPT-4o achieved satisfactory results in the few-shot setting. This highlights that fine-tuning, as implemented in CAEF, is crucial for enabling the effectiveness of multi-query approaches.
>
> \[1] Cai, Weilin, et al. "A Survey on Mixture of Experts." *arXiv preprint arXiv:2407.06204* (2024).
>
> \[2] Lee, Nayoung, et al. "Teaching arithmetic to small transformers." *arXiv preprint arXiv:2307.03381* (2023).

---

### Author Response · Authors · 2024-11-26

We sincerely appreciate the valuable suggestions provided by all the reviewers. Due to time constraints, we have revised and resubmitted an updated version, with a focus on improving the textual content. Further revisions will follow, including enhancing the clarity of the figures and refining the overall structure of the paper, to ensure that CAEF can be more easily understood by readers.

---

### Meta-Review · Area_Chair_NbbJ · 2024-12-20

**Metareview:**

This paper introduces the Composable Arithmetic Execution Framework (CAEF), an approach to enhance Large Language Model (LLM) arithmetic capabilities. CAEF simulates a Turing machine within an LLM, breaking down computations into sequential <state, command> pairs processed by specialized LoRA-adapted modules for alignment and execution. The design enables more accurate calculations, especially for large numbers, outperforming both fine-tuned LLaMA and GPT-4o in certain scenarios. This method is a good proof of concept, with well designed experimental results.

However, CAEF suffers from several limitations. The most significant is its inefficiency, requiring multiple LLM queries per computation step and resulting in computational complexity that increases with operand length. This approach is also limited because it does not utilize external tools and cannot automatically select its operation LoRA adapters. There are also questions from reviewers about the robustness of the approach, especially given particular failure modes based on digit repetition and the potential for cascading errors. Though the authors address some of these concerns with further experimentation, the core issues around practicality remain.

Overall, this paper presents an interesting methodology for LLM arithmetic computation, demonstrating improved accuracy for complex arithmetic tasks. While the theoretical concept is sound and the experimental results are promising, the current implementation is not practical due to efficiency constraints and lack of flexibility. It could be strengthened by increased clarity regarding the execution process, a more detailed computational complexity analysis, and further exploration of the identified limitations and error analysis.

**Additional Comments On Reviewer Discussion:**

Reviewers first questioned CAEF's efficiency (sUkz), complexity and non-generic adapter selection (CdN9), the need for LLM Turing machine simulation (WNj4), and the clarity and motivation for the approach (F4pL). Authors responded by clarifying execution, analyzing complexity, adding CoT baselines, addressing modularity, and discussing the non-practical nature of the method, and removing anthropomorphism. While the rebuttals improved clarity and methodology, core limitations around efficiency and practicality remained. This resulted in mixed changes to reviewer scores with F4pL increasing and sUkz decreasing.

---

### Decision · Program_Chairs · 2025-01-22

Reject